

# Penetration of interferometric radar signals in Antarctic snow

Helmut Rott[1,2] *, Stefan Scheiblauer[1], Jan Wuite[1], Lukas Krieger[3], Dana Floricioiu[3], Paola Rizzoli[4], Ludivine Libert[1], Thomas Nagler[1]

[1] ENVEO IT GmbH, Innsbruck, Austria
[2] Department of Atmospheric and Cryospheric Sciences, Univ. of Innsbruck, Innsbruck, Austria
[3] Remote Sensing Technology Institute, DLR, Oberpfaffenhofen, Germany
[4] Microwaves and Radar Institute, DLR, Oberpfaffenhofen, Germany

* Correspondence to: Helmut Rott (helmut.rott@enveo.at)

**Abstract.** Synthetic aperture radar interferometry (InSAR) is an efficient technique for mapping the surface elevation and its temporal change over glaciers and ice sheets. However, due to the penetration of the SAR signal into snow and ice the apparent elevation in uncorrected InSAR digital elevation models (DEMs) is displaced versus the actual surface. We studied relations between interferometric radar signals and physical snow properties and tested procedures for correcting the elevation bias. The work is based on satellite and in-situ data over Union Glacier in the Ellsworth Mountains, West Antarctica, including interferometric data of the TanDEM-X mission, topographic data from optical satellite sensors and field measurements on snow structure and stratigraphy undertaken in December 2016. The study area comprises ice-free surfaces, bare ice, dry snow and firn with a variety of structural features related to local differences in wind exposure and snow accumulation. Time series of laser measurements of NASA's Ice, Cloud and land Elevation Satellite (ICESat) and ICESat-2 show steady state surface topography. For area-wide elevation reference we use the Reference Elevation Model of Antarctica (REMA). The different elevation data are vertically co-registered on a blue ice area and an ice-free slope, surfaces not affected by radar signal penetration. The backscatter simulations with a multi-layer radiative transfer model show large variations for scattering of individual snow layers due to different size and structure of the scattering elements. The average depth-dependent backscatter contributions can be approximated by an exponential function. We obtain estimates of the elevation bias by inverting the interferometric volume correlation coefficient (coherence) applying a uniform volume model for describing the vertical loss function. Whereas the mean values of the computed elevation bias and the elevation difference between the TDM DEMs and the REMA show good agreement, a trend towards overestimation of penetration is evident for heavily wind-exposed areas and towards underestimation for areas with higher accumulation rates. The angular gradients of the backscatter intensity show also distinct differences between these two domains. This behaviour can be attributed to the anisotropy of the snow/firn volume structure showing differences in the size and shape of the scattering elements and in stratification related to snow accumulation and wind-driven erosion and deposition.

## 1. Introduction

Digital elevation models (DEMs) derived from across-track interferometric synthetic aperture radar (InSAR) data are a main data source for mapping the surface elevation and its temporal change over glaciers and ice sheets. Single-pass (SP) InSAR systems, such as the TanDEM-X (TDM) mission, are of particular interest for this task as they are not affected by variations in the atmospheric phase delay, ice motion and temporal decorrelation. For the analysis and interpretation of InSAR elevation over snow and ice the effects of signal penetration have to be taken into account. The surface inferred from uncorrected InSAR elevation data refers to



the position of the scattering phase centre in the snow/firn medium, resulting in an elevation bias versus the actual surface (Dall, 2007). The position and strength of scattering sources in the snow volume and the absorption and scattering losses are main factors defining the depth of the phase centre below the snow surface. Backscatter contributions from sources in different depths within a volume scattering medium, observed under slightly different incidence angles, are causing a spectral wavenumber shift and decorrelation (Gatelli et al., 1994).

Weber Hoen and Zebker (2000, 2001) derived a formulation for estimating the power-penetration depth, $d_p$, in dry snow from the interferometric coherence, applying a radiative transfer model for estimating spatial decorrelation in a volume of uniformly distributed and uncorrelated scatterers characterized by exponential extinction. They applied this formulation to derive the C-band penetration depth for different sites in Greenland from the coherence of 3-day repeat-pass InSAR data of the ERS-1 SAR mission. Forsberg et al. (2000) and Dall et al. (2001) compared surface elevation measured by airborne laser altimetry and C-band single-pass SAR interferometry on the Geiki ice cap in Greenland. They report zero InSAR elevation bias for wet snow and an average bias of about 10 m for dry snow and firn. Dall (2007) studied relations between the InSAR elevation bias and the power penetration depth in uniform volumes. He shows that the depth of the mean phase centre in a volume scattering medium is approximately equal to the two-way penetration depth, $d_{p2}$, if the latter is smaller than about 10% of the height of ambiguity ($H_a$), the height difference for a phase shift of $2\pi$. Fischer et al. (2019a; 2019b; 2020) studied various concepts for characterizing and modelling the vertical backscatter distribution and retrieving the InSAR penetration bias in the percolation zone of Greenland based on airborne polarimetric multi-baseline InSAR data and in situ measurements of snow structural properties.

In recent years SP-InSAR data of the TDM mission were widely applied for mapping surface elevation and elevation change on glaciers and ice sheets. The TDM mission employs a bi-static interferometric configuration of the two satellites TerraSAR-X and TanDEM-X flying in close formation in order to form a single-pass SAR interferometer (Krieger et al., 2013). Rizzoli et al. (2017a) compared surface elevation over Greenland measured by NASA's Ice, Cloud and land Elevation Satellite (ICESat) laser altimeter with the TanDEM-X global digital elevation model (DEM). They report for frozen snow and firn in the wet snow zone, the lower and upper percolation zone, and the dry snow zone mean values of the X-band InSAR penetration bias of 3.7 m, 3.9 m, 4.7 m, and 5.4 m, respectively. Abdullahi et al. (2019) use a linear regression model for estimating the elevation bias in TDM DEMs of northern Greenland. The model is based on empirical relations between coherence and backscatter intensity with the difference between the uncorrected TDM DEM and airborne laser-altimeter surface heights.

The complex layered structure of polar snow and firn has a major impact on radar signal propagation and interferometric coherence, an obstacle for establishing a generally applicable, physically-based method for estimating the elevation bias of InSAR products. The work presented in this paper takes on this open issue, exploring relations between interferometric parameters and physical snow properties and investigating the feasibility of deducing the elevation bias from the interferometric correlation. The study is based on interferometric data of the TDM mission, data from optical satellite sensors and field measurements undertaken in December 2016 on Union Glacier in the Ellsworth Mountains, Antarctica. Logistic support was provided by the private company Antarctic Logistics & Expeditions LLC (ALE) which conducts aircraft flights from Punta Arenas, Chile, to Union Glacier and operates in summer a well-equipped field station and logistic facilities. The



study area comprises ice-free surfaces, bare ice and dry snow and firn exhibiting a diversity of structural features
attributed to local differences in wind exposure and snow accumulation. Time series of ICESat laser
measurements from 2003 to 2009 and ICESat-2 data show near steady state surface topography, facilitating the
intercomparison of TDM and optical elevation data.

In Sect. 2 we describe the study area, present details on the satellite data base and give an account on the
structure and morphology of snow and firn at different sites. Sect. 3 explains the approach for the vertical co-
registration of the different DEMs. Furthermore, checks on the temporal stability of surface elevation are
presented based on altimetric time series. Sect. 4 reports on the vertical backscatter distribution in snow/firn
volumes and presents the theoretical relation between volumetric coherence and the InSAR elevation bias. Sect.
5 shows the spatial patterns of backscattering, coherence and the elevation bias of the different TDM scenes and
explains the interdependencies between these parameters. Sect. 6 deals with the inversion of the volumetric
coherence in terms of the InSAR elevation bias and compares the retrieved bias with elevation differences
between TDM DEMs and optical data. In Sect. 7 we discuss merits and constraints of the inversion approach in
the context of the study results and address open issues.

## 2.   Study area and data

Union Glacier flows from the ice divide in the Heritage Range, Ellsworth Mountains, down to the Constellation
Inlet on Ronne Ice Shelf. The glacier section immediately downstream of the main mountain range is exposed to
strong katabatic winds so that bare ice appears on the surface (Fig. 1). The blue ice area (BIA) has a negative
specific surface mass balance in the order of several centimetres per year due to sublimation (Rivera et al.,
2014). On the BIA an ice runway for landing heavy airplanes on wheels is maintained from November to March.
The ALE camp is located 8 km downstream of the ice runway (near P3 in Fig. 1).

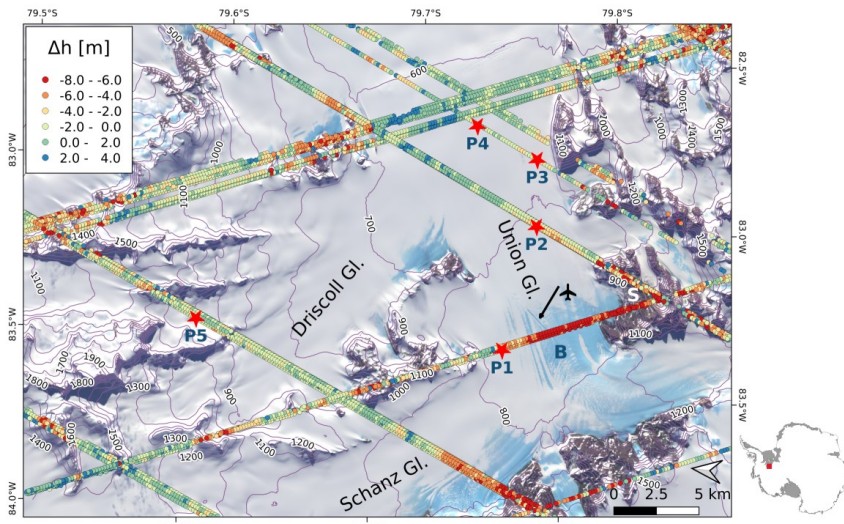


**Figure 1.** Landsat-8 image acquired on 6 December 2016 (composite of bands 5, 4, 2) with ICESat tracks. Points: elevation
difference Δh (ICESat minus TDM global DEM), colour coded from Δh = -8 m to +4 m. P1 to P5: Locations of snow pits. B
– blue ice area., S – ice free slope. The arrow points to the landing strip.



GPS measurements at stakes, performed during the period 2007 to 2011, show ice velocities in the order of 20 m

$a^{-1}$ at the glacier gate across the runway (Rivera et al., 2010; 2014). For 2008 to 2012 a mean wind speed of 16.3 knots with predominant direction from south-west was measured at an automatic station close to the runway. Wind speed and direction are very consistent. Rivera et al. (2014) report a mean specific mass balance, $b_n$, of - 0.10 m w.e. $a^{-1}$ measured at 29 stakes on the BIA during 2007 to 2011. The intensity of the katabatic winds declines downstream of the BIA so that snow accumulates and the surface mass balance is positive.

Accumulation measurements at 11 stakes up to 15 km downstream of the BIA show a maximum $b_n$ of 0.20 m w.e. $a^{-1}$ at a stake near the ALE camp (Rivera et al., 2014). Hoffmann et al. (2020) collected and analysed six shallow ice cores in the wider Union Glacier region. One of the cores was drilled on Union Glacier itself, about 2.5 km west of P3, showing for 1989 – 2013 a mean $b_n$ of 0.18 m w.e. $a^{-1}$.

Differences in exposure to wind are a main factor for local variations in the accumulation rate and in the

structural properties of snow and firn. This is evident in differences of the microstructure and stratigraphy observed in snow pits, ranging from coarse-grained dense snow with wind-crusts near the runway (pit P1), located in the main pathway of the katabatic wind, to finer-grained and softer snow at P5 on a lateral slope of Driscoll Glacier.

Uribe et al. (2014) operated two radar sensors during an oversnow campaign in December 2010, measuring the

total ice thickness and the thickness and structure of the firn layers along an 82 km track, starting on Union Glacier and proceeding along Driscoll and Schanz glaciers up to the Ellsworth Plateau. The total thickness of the firn layer varies significantly along this track, even within short distances. For example, a radargram of a 6 km transect extending from the confluence with Driscoll Glacier across Union Glacier towards the camp shows thickness values of the snow/firn layer ranging from zero on blue ice to a maximum of 34 m close to the camp.

**2.1 TanDEM-X data**

The TanDEM-X data for this study comprise one tile of the TDM global (TDMgl) DEM and raw SAR data from several dates for compiling topography, backscatter intensity and coherence products. Tile TDM1_DEM_04_S80W084_V01_C of the global DEM is used, extending from 79° to 80° S and 82° to 84° W and referring to the coordinate reference system WGS84-G1150. This tile was obtained by mosaicking multiple

single DEM scenes acquired between 6 May 2013 and 23 August 2014. The pixel spacing is 0.4 arcsec in northing and 1.2 arcsec in easting, corresponding to 12.4 m x 6.5 m at 80° latitude. For the TDMgl elevation over ice sheets penetration corrections were applied, using ICESat data as elevation reference (Wessel et al., 2016; Rizzoli et al., 2017b). For Antarctica (excluding coastal regions) a mean penetration bias was derived for each of eleven extended homogeneous areas (fixed blocks) located in different sections of the ice sheet. For all

other areas the elevation is adjusted by spatial interpolation between these blocks, regionally applying bulk values that are not accounting for different surface types.

For producing DEMs from raw bistatic SAR data of individual tracks (so-called Raw DEMs) we used the operational Integrated TanDEM-X Processor (ITP) of the German Aerospace Center (DLR) (Rossi et al., 2012). The Raw DEM pixel spacing is 6 m x 3 m. Complementary to each Raw DEM the ITP provides maps of the

height error, the SAR amplitude, the backscattering coefficient and the interferometric coherence, as well as a flag mask indicating critical areas. The Height Error Map (HEM) delivers the height errors for each DEM pixel caused by random noise. This error is largely driven by the phase uncertainty for which the coherence is the





main factor. Low pass filtering is an efficient means for reducing the random height error. The HEM map for the TDMgl DEM of the study region shows over flat terrain and gentle slopes random height errors (standard deviation) ranging from 0.3 m to 1.2 m.

Specifications of the TDM data used in this study are listed in Table 1. We selected scenes with different incidence angles and baselines in order to check the impact of these parameters on coherence, backscatter intensity and signal penetration. According to the HEM maps, the random errors for the Raw DEMs, excluding steep slopes, range from 0.7 m to 3.0 m. The spatial variations can mainly be attributed to phase noise related to thermal and volume decorrelation. For the estimation of signal penetration we use averages over multiple pixel windows in order to reduce the uncertainty.

**Table 1.** Specifications of TanDEM-X data used for DEM production and generation of backscatter and coherence images. $\theta_i$ is the incidence angle in the scene centre. $B_n$ is the effective interferometric baseline, $H_a$ is the height of ambiguity, $k_{zVol}$ is the vertical interferometric wavenumber in the snow volume assuming a density of 400 kg m$^{-3}$. SAR operation mode: bistatic.

| Label | Date | Rel. Orbit / Scene | Look direction | Polarisation | $\theta_i$ [deg] | $B_n$ [m] | $H_a$ [m] | $k_{zVol}$ [rad m$^{-1}$] |
|-------|------|-------------------|----------------|--------------|------------------|-----------|-----------|---------------------------|
| T2013A | 2013-05-06 | 105 / 15 | Left | HH | 40.9 | 107.4 | -65.6 | 0.111 |
| T2013B | 2013-05-22 | 198 / 14 | Left | HH | 38.6 | 106.5 | -61.2 | 0.121 |
| T2014A | 2014-05-09 | 14 / 7 | Left | HH | 37.5 | 145.8 | -42.9 | 0.173 |
| T2014B | 2014-06-12 | 233 / 35 | Left | HH | 40.8 | 123.5 | -56.6 | 0.128 |
| T2016 | 2016-12-10 | 18 / 2 | Right | HH & VV | 21.6 | 50.0 | -67.3 | 0.120 |
| T2018 | 2018-01-10 | 18 / 2 | Right | HH & VV | 22.1 | 30.2 | -112.0 | 0.072 |

### 2.2 Topographic data from optical satellite sensors

Topographic data from the ICESat and ICESat-2 missions and the Reference Elevation Model of Antarctica (REMA), derived from very high resolution optical stereo images (Howat et al., 2019), are available as reference for estimating the elevation bias in the InSAR DEMs. The study area is covered by several tracks of the ICESat and ICESat-2 altimeters. Elevation data were acquired by ICESat during several campaigns between April 2003 and October 2009. We use GLAH12 GLAS/ICESat L2 Global Antarctic and Greenland Ice Sheet Altimetry Data (HDF5), Release 34 (Zwally et al., 2014). This product provides geolocated and time tagged surface elevation estimates, referenced to the TOPEX/Poseidon ellipsoid, corrected for atmospheric delays and tides. The laser footprint size is 60 m to 70 m, the distance between the footprint centres is approximately 170 m. The analysis of repeat-track data allows the detection of the surface elevation change after correcting for elevation differences caused by horizontal shifts of individual footprints. A main cause for the height error of ICESat footprints is the uncertainty in beam pointing, causing slope-induced errors (Brenner et al., 2007; Zwally et al., 2011). Co-located ICESat footprints of 18 campaigns on level terrain in East Antarctica show intercampaign elevation biases between -3.6 cm and +14.7 cm, confirming the high temporal stability (Hofton et al., 2013).

Regarding ICESat-2 we use ATLAS/ICESat-2 L3A Land Ice Height, Version 2, Land Ice Along-Track Height Product (ATL06), from the time span 2018-10-14 to 2019-09-01. This data set provides geolocated land-ice surface heights above the WGS 84 ellipsoid, ITRF2014 reference frame, and ancillary parameters including error estimates and quality flags (Smith et al., 2019a). ATL06 heights represent the mean surface height averaged along 40 m segments of ground track, 20 m apart, for each of the six beams of the Advanced Topographic Laser



Altimeter System (ATLAS) instrument on board the ICESat-2 observatory. The land-ice height is defined as estimated surface height of the segment centre for each reference point, using median-based statistics (Smith et al., 2019b).

For spatially detailed comparisons of elevation we use the REMA DEM Tile Nr. 32-19 with 8 m posting, covering Union Glacier (Howat et al., 2019). The dates of the image acquisitions for this tile range from January
2014 to December 2015. The absolute height is based on vertical registration to CryoSat-2 altimetry data, acquired in SARIn mode. In order to account for the CryoSat signal penetration a uniform value of 0.39 m was added to the CryoSat-2 co-registered heights over the Union Glacier region, regardless of the surface type (Howat et al., 2019). This needs to be taken into account for using the REMA data as elevation reference because the study area includes bare ground, ice surfaces, and snow and firn with different structural properties
affecting radar signal penetration. The vertical error estimates for REMA in the region of interest range from 1.0 m to 1.4 m. The error value is the standard error from the residuals of the registration to altimetry. The error due to the use of the bulk Cryosat-2 based penetration correction is not included in this error estimate.

**2.3 Snow pit measurements**

For the snow pits we selected sites covered by ICESat footprints that show different values of coherence and
backscatter intensity in TDM data. Backscatter properties of dry snow and firn are controlled by snow microstructure which is also a main factor for X-band radar signal penetration. In the study region the impact of melt for the snow metamorphism is marginal. We detected evidence for melt events in two of the five snow pits: a thin ice crust in 1.1 m depth at P5 and two thin ice crusts along with one ice layer of 4 cm thickness at P1. The temperature record from March 2010 to February 2014 at the meteorological station near the runway shows a
mean annual air temperature of -21.1 °C and mean monthly temperatures of -8.6 °C for December and -9.3 °C for January. During those years a few short events with air temperatures close to the melting point were recorded.

Profiles of snow density, temperature, hardness, grain size and shape are shown in Fig. 2. The pits vary in depth between 1.6 m and 2.3 m. The observed grain size refers to the maximum axis length of prevailing grains.
Hardness was estimated by the hand test, ranging from very low (R1) to very high (R5) for snow and R6 for ice. The mean density of snow/firn for layers of 0.5 m vertical extent is specified in Table 2. Grain size and hardness show significant differences between the five measurement sites. The size and shape of the snow grains and the sequence and properties of snow/firn layers are arising from accumulation history, exchange processes of radiation, turbulent heat and mass at the snow/air interface, and vapour diffusion in the snow volume. Down to
about 2 m depth the temperature gradient metamorphism is the dominating process for grain growth, triggered by seasonal temperature variations (Alley, 1988; Colbeck, 1983). Average temperature gradients in the top metre of the five snow pits were in the order of 10 °C m$^{-1}$. At lower depth equi-temperature metamorphism takes over as dominant process for grain growth. Differences in the average grain size of the pits can, at least partly, be attributed to the snow age following from different accumulation rates. Courville et al. (2007) studied the
microstructure of snow and firn in a megadune region in East Antarctica. They show that local differences in grain size, thermal conductivity, and permeability are related to spatial accumulation variability in which already relatively small differences in the accumulation rate due to wind redistribution cause significant differences of physical snow properties.

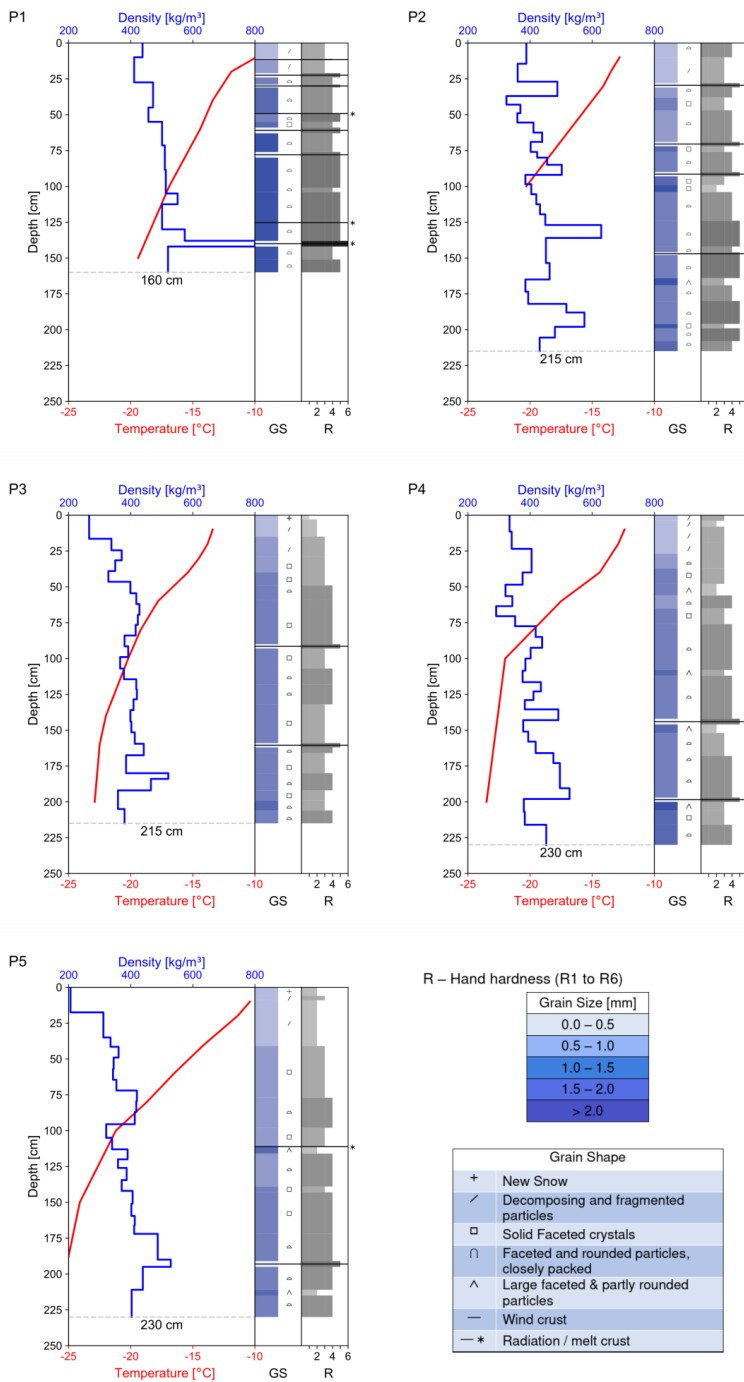

**Figure 2.** Vertical profiles of snow temperature, density, grain size (GS), grain shape and hand hardness (R) for snow pits P1 to P5 on Union Glacier, December 2016. The grain size refers to the maximum axis length of the prevailing snow grains.

**Table 2.** Mean density of snow/firn for layers of 0.5 m vertical extent of snow pits P1 to P5 on Union Glacier. The snow pit altitude refers to the REMA DEM.

|  | P1 | P2 | P3 | P4 | P5 |
|---|---|---|---|---|---|
| Altitude [m] | 756.8 | 690.1 | 674.1 | 656.0 | 1133.3 |
| Depth | Snow density [kg m$^{-3}$] | | | | |
| 0 to 0.5 m | 443 | 390 | 323 | 366 | 286 |
| 0.5 to 1.0 m | 499 | 422 | 408 | 369 | 372 |
| 1.0 to 1.5 m | 548 | 471 | 399 | 408 | 371 |
| 1.5 to 2.0 m |  | 467 | 419 | 472 | 451 |

Snow pit P1 exhibits the largest grains, the highest snow density, thin ice layers and wind crusts. Accumulation data are not available, but from the closeness to the BIA it can be concluded that the mean accumulation rate is well below the accumulation rate near the ALE camp. Due to the high exposure to katabatic winds the stratification does not allow a clear identification of annual accumulation layers. In some years sublimation and wind erosion may result in negative mass balance. The higher hardness values compared to the other sites can be attributed to more frequent exposure to high wind speeds, the erosion and deposition of blowing snow and greater age due to low accumulation. Two thin ice crusts (5 mm thickness) at 0.49 m and 1.25 m depth are possibly tracing back to radiation penetration causing melt below the frozen surface (Colbeck, 1989). An ice layer of 4 cm thickness between 1.38 and 1.42 m depth, with air bubbles of up to 2 mm size, indicates an intensive melt event.

P2 is the snow pit with the highest average snow density next to P1. It is located half-way between the runway and the ALE camp, more exposed to katabatic winds than the camp so that the average accumulation rate should be lower than at P3 and P4. The stratigraphy down to 2 m depth shows four layers of high density with comparatively fine-grained snow, typical for wind packs, and several thin wind crusts. Softer layers with faceted grains show up below wind packs, but a clear assignment to seasonal or annual layers is not possible.

The pits P3 and P4, located in the vicinity of the camp, show lower mean density and less variations of density with depth. P4 is located slightly upstream of stake B10 for which Rivera et al. (2014) report a specific mass balance $b_n = 0.17$ m w.e. a$^{-1}$ for 2008-2009. Down to the depth of 2.04 m the P4 stratification shows four, comparatively thick, hard layers with softer, large-grained snow below. The total snow mass down to 2.04 m amounts to 0.80 m w.e. Assuming that the transitions from hard to soft layers corresponds to late summer horizons and accounting for the lack of two months to cover the full 4-year period implies an annual accumulation rate $b_n = 0.21$ m w.e. a$^{-1}$. At P3 the sequence of layers is less distinct. This site is located in 300 m cross-wind distance of the camp and may be affected by local perturbations of snow drift during summer when the camp is set up in full extent.

P5 is located at 1133 m elevation on a slanting lateral branch of Driscoll Glacier that extends uphill towards the Pioneer Heights, 400 m in altitude above the confluence with Union Glacier. The site is not exposed to the strong katabatic winds that are blowing along the main branch of Union Glacier. The grain size is smaller and the snow is softer than at the other sites. A melt crust of 3 mm thickness was found in 1.14 m depth, most likely related to a short event with comparatively warm temperatures on 17-18 January 2016. The snow mass above this crust amounts to 0.38 m w.e. A thin hard layer in 2.11 m depth with a soft, coarse-grained layer below refers probably to the 2015 late–summer horizon. The snow mass between the wind crust in late summer 2015 and the



melt crust in January 2016, a period of about 13 months, amounts to 0.41 m w.e. These two accumulation
estimates indicate for this site about twice the accumulation rate of the main glacier near the ALE camp.

### 3.   Topographic reference data for the penetration analysis

The apparent glacier surface in an InSAR DEM refers to the position of the scattering phase centre in the snow
and firn volume. The elevation bias, $h_b$, is the difference between the apparent elevation derived by means of the
InSAR method, $h_{insar}$, and the true surface elevation, $h_s$:

$$h_b = h_{insar} - h_s .$$    (1)

For obtaining reliable estimates of the elevation differences between DEMs from different sources precise
vertical co-registration on stable surfaces is needed (Nuth and Kääb, 2011). For studying the penetration-related
elevation bias we co-register the TDM DEMs on surfaces devoid of penetration with elevation data of optical
sensors. On these surfaces the TDM DEMs show vertical offsets up to some metres. We use the notation Δh for
specifying the elevation difference between optical data and the un-registered TDM DEMs on surface scattering
targets:

$$\Delta h = h_{optical} - h_{TDM,unreg}$$    (2)

Suitable targets for vertical co-registration in the study area are the BIA and bare ground on an ice-free slope
bordering the BIA ("S" in Fig. 1). The slope has a mean inclination of about 16 degrees and contains sections of
varying steepness. On slopes horizontal shifts between pixels of topographic data to be co-registered cause
slope-dependent elevation biases, in particular if the data are from sensors with different observation geometries
and spatial resolution (Nuth and Kääb, 2011). Therefore we use data from moderately inclined slope sections and
from the BIA (with a mean inclination of about one degree) for quantifying the vertical offset between TDM and
optical elevation data.

We use the notation dh for denoting the elevation difference between the TDM DEMs and optical elevation data,
vertically co-registered on surface scattering targets:

$$dh = h_{TDM} - h_{optical}$$    (3)

In case of temporal coincidence or stable topography dh corresponds to the interferometric elevation bias.
Though the time series of ICESat data indicate temporal stability of surface elevation, minor errors due to
temporal changes in elevation cannot be fully excluded.

### 3.1 Temporal stability of surface elevation

Because of the lack in temporal coincidence between the TDM and optical elevation data we checked the
temporal variability using ICESat time series. The main section of the BIA was crossed by ICESat repeat tracks
on seven dates between May 2004 and November 2009 (Fig. 1). The mean difference in elevation Δh between
the ICESat footprints and the corresponding TDMgl cells (mean values of 5 x 5 pixels) is -6.76 m. The standard
deviation for the 126 samples of the time series is 0.43 m (Table S1 in the Supplement). The mean Δh values
between the individual ICESat tracks on the different dates and TDMgl range from -6.61 m to -6.86 m without
any distinct temporal trend, indicating high temporal stability. The stability of surface elevation on the BIA is

also confirmed by the GPS time series of Rivera et al. (2014). The $\Delta h$ value of -6.76 m can mainly be attributed to the bulk penetration correction of the TDMgl DEM. The ICESat-2 (IC2) data set for the BIA includes eight tracks with altogether 345 spots, extending along the eastern and western margins of the BIA which are occasionally covered by snow. The mean elevation difference and standard deviation are: $\Delta h$(IC2-TDMgl) = -6.99 m, $\sigma_{\Delta h}$ = 0.38 m.

On the ice-free slope the variance of $\Delta h$ is higher than on the BIA. The different footprint sizes and observation geometries of the ICESat and TDM sensors are prone to slope-induced errors for DEM differencing. In order to avoid steep slope sections we excluded for the comparison all cells of 5 x 5 TDM pixels with a standard deviation of elevation larger than 5 m. Under this constraint only 21 ICESat pixels of the whole time series qualify for the comparison on the slope, yielding a mean $\Delta h$ of -6.91 m and $\sigma_{\Delta h}$ of 0.84 m.

Another ICESat time series for checking the temporal behaviour of surface elevation extends across the main glacier near P4 where the average elevation bias of TDMgl due to penetration is several metres. The ICESat data set comprises seven closely-spaced tracks acquired between 11 April 2003 and 12 February 2008. The mean value and standard deviation of the elevation difference between ICESat and un-registered TDMgl on the central section of the glacier are: $\Delta h$ = 0.09 m, $\sigma_h$ = 0.40 m (Table S2). The mean $\Delta h$ values on individual dates range

from -0.03 m to 0.21 m without any obvious temporal trend.

### 3.2 Vertical co-registration of the DEMs

The use of the REMA elevation data as reference enables spatially detailed estimates of the interferometric elevation bias. The mean value and standard deviation of the elevation difference between ICESat and REMA over the BIA are: $\Delta h$ = -0.33 m, $\sigma_{\Delta h}$ = 0.38 m (Table S1). This value differs by 6 cm from the bulk penetration

correction (-0.39 m) that was applied to CryoSat-2 elevation over the Union Glacier area in order to obtain the absolute height reference for the REMA DEM. This correction introduces a bias because the actual CryoSat-2 signal over bare ice is dominated by surface return whereas the correction accounts for penetration in dry snow. The difference between ICESat-2 and REMA on the BIA is: $\Delta h$ = -0.54 m, $\sigma_{\Delta h}$ = 0.46 m. On the ice-free slope the elevation differences of REMA versus ICESat, ICESat-2 and TDMgl elevation data show high standard

deviations. Therefore we use the BIA as reference site for vertical co-registration between the TDM DEMs and REMA.

For cross-comparing the TDM and REMA elevation data we outlined an area of 5 km² extent in the central section of the BIA that is crossed by the ICESat tracks. The mean elevation difference $\Delta h$ between REMA and TDMgl is -6.37 m, the standard deviation at 8 m x 8 m pixel size is 0.62 m. We use the value of -6.37 m for

vertical co-registration of the TDMgl DEM. Data of the same polygon are used for vertical co-registration of the other TDM DEMs which as un-registered DEMs show vertical shifts vs. REMA of 2 m to 3 m.

### 4.    Vertical backscatter distribution and volumetric coherence

For inverting the observed interferometric coherence in terms of the elevation bias the pre-setting of a vertical profile for scattering in the snow/firn volume is required because it is not possible to derive the penetration

properties of individual layers. We tested the applicability of the uniform volume approach for describing the





observed backscatter intensity and show the resulting relation between the volumetric coherence and the interferometric elevation bias.

### 4.1 Representation of the vertical backscatter profile

Common approaches for relating the penetration depth, $d_p$, in a frozen snow/firn medium to the observed coherence are based on the assumption of exponential extinction in a uniform volume (Weber Hoen and Zebker, 2000; Dall, 2007; Fischer et al., 2020). The power received from depth z below the surface in a homogeneous lossy medium is described by an exponential function:

$$P_r(z) = P_{tot}\ exp\left[\frac{2\,z\,k_e}{\cos\theta_r}\right] = P_{tot}\ exp\left[\frac{2\,z}{d_p}\right] \tag{4}$$

$P_{tot}$ is the total backscattered power. $P_r$ is the backscattered power from depth z that is attenuated due to absorption and scattering in the layers above. In the radiative transfer (RT) approach for single scattering the extinction coefficient, $k_e$, accounts for absorption and scattering losses:

$$k_e = k_a + k_s \tag{5}$$

where $k_a$ and $k_s$ [m$^{-1}$] are the absorption and the scattering coefficients per unit volume. In a uniform volume the one-way power penetration length $d_l$ [m], where the intensity of the signal is attenuated to 37% of the incident signal, is given by

$$d_l = \frac{1}{k_e} = \frac{1}{k_s + k_a} \tag{6}$$

The one-way power penetration depth referring to vertical direction is obtained by accounting for the refraction angle $\theta_r$:

$$d_p = d_l \cos\theta_r\ . \tag{7}$$

If the elevation bias is much smaller than the height of ambiguity the 2-way power penetration depth, $d_{p2}$, can be used as reference for the location of the scattering phase centre and as estimate for the interferometric elevation bias (Dall, 2007).

The approach described in Eq. 4 implies constant scattering and extinction coefficients. Actual snow/firn volumes exhibit vertical variations in scattering properties related to layers of different microstructure and density and show a general trend of densification with depth (Fig. 2). In order to check the suitability of the exponential function for inferring the penetration bias from the observed coherence we performed backscatter simulations using the multilayer Snow Microwave Radiative Transfer (SMRT) thermal emission and backscatter model of Picard et al. (2018). The SMRT offers the choice of different electromagnetic and microstructure models for computing the scattering and absorption coefficients and the scattering phase function in a given layer. We apply the sticky hard sphere (SHS) model for characterizing the microstructure and the improved Born approximation (IBA) for computing volume scattering and absorption.

Input parameters for describing the microstructure of each layer with the SHS model are the snow density, the temperature, the diameter of the prevailing snow grains and the stickiness. Closely packed ice particles have a tendency to form clusters and bonds. The stickiness parameter accounts for sintering and clustering of snow grains, forming aggregates that are larger than individual grains. The collective scattering and wave interaction effects of the aggregates result in increased scattering compared to individual grains and show a different phase



function with more forward scattering. Löwe and Picard (2015) found stickiness to be an essential parameter when modelling snow as a sphere assembly. They show that the stickiness parameter can be objectively estimated from micro-tomography images. However, objective methods deriving the stickiness parameter from field observations are still pending.

Whereas the stickiness parameter accounts for increased scattering due to sintering, the close packing of particles introduces near field interactions causing reduced scattering. Compared to the assumption of independent scattering elements the scattering in a dense medium decreases with increasing volume fraction of the scatterers larger than about 0.2. Dense Medium Radiative Transfer (DMRT) models, including the IBA, account for dense media effects. We performed test runs with the DMRT model with the Quasi-Crystalline Approximation (QCA) of Mie Scattering (DMRT-QMS) of Tsang et al. (2007) and Chang et al. (2014), showing similar results as the IBA-SHS approach.

For the backscatter simulations of the snow pit sites we accounted for 25 layers with different physical properties. Down to the bottom of the snow pits the grain size and density data are based on the field measurements. The snow density below is adopted from the density profile of the firn core GUPA-1 of Hoffmann et al. (2020), located near P3. For grain size we use the maximum axis length of the prevailing snow grains in each snow layer and assume a moderate increase of grain size with depth below 2 m. This assumption takes into account observations of microstructure in snow pit and firn-core samples down to 16 m depth on the West Antarctic Plateau showing significant increase of the mean grain intercept length in the top 2 metres whereas below the increase with depth is weak (Rick and Albert, 2004).

In the SHS model the stickiness parameter, $\tau$, accounts for the degree of bonding between spherical particles. $\tau = 0$ corresponds to infinite stickiness and $\tau = \infty$ to the nonsticky case (Tsang et al., 2013). Typical $\tau$-values for computing X-band backscatter range from $\tau = 0.1$ for coarse metamorphic snow to $\tau = 0.5$ for fine-grained snow. The computed scattering coefficient, $k_s$, of rounded grains with $\tau = 0.5$ is only slightly larger than then nonsticky case, whereas with $\tau = 0.1$ the scattering coefficient is larger by about one order of magnitude. DMRT-QMS backscatter simulations using $\tau$-values between 0.12 and 0.35 for snow layers of different metamorphic state show good results compared to the measured $\sigma°$ of seasonal snowpacks in northern Finland (Chang et al., 2014). For comparatively shallow metamorphic snow in Colorado $\tau = 0.1$ is used for matching observations and measurements (Tsang et al., 2013). We chose $\tau$-values in order to match the computed and observed TDM backscatter coefficients and the 2-way power penetration depth deduced from the height difference (dh) between the TDM DEMs and the REMA. The $\tau$-values range from $\tau = 0.1$ for layers with large grains and clusters to $\tau = 0.2$ for near surface layers.

Fig. 3 shows X-band HH-polarized backscatter simulation for 40° incidence angle referring to snow/firn properties of Pit 2 and Pit 4. Pit 2 is located 4 km downstream of the BIA in an area of high wind exposure, exhibiting coarse-grained, clustered snow and distinct differences in density between individual layers. Pit 4 is located in a zone of reduced wind force and higher accumulation rate, featuring smaller average grain size and less variations of density with depth. The computed backscatter coefficient $\sigma°$(HH) is -7.96 dB for Pit 2 and -11.81 dB for Pit 4. Both values differ by less than 0.3 dB from the mean $\sigma°$ of the 2013 and 2014 TDM scenes (Table S3). Based on the computed vertical backscatter profile the 2-way power penetration depth amounts to 4.72 m at Pit 2 and 7.25 m at Pit 4, close to the mean differences dh of the respective TDM 2013/2014 and





REMA elevations (Table S4). The computed one-way penetration length at Pit 2 is 10.85 m, close to X-band values (10.4 m) measured on the East Antarctic Plateau (Rott et al., 1993).

The contributions of the individual layers to the total observed power vary with the scattering properties related to the microstructure. There is a general decrease with depth due to the attenuation in the layers above and due to

the depth-dependent firn compaction (Figs. 3a and 3c). Layers with coarse grains and grain clusters show higher backscatter coefficients but are thinner than compact layers of higher density such as wind slabs. The variations between individual layers with different scattering properties are smoothed out in the depth-dependent function of the total backscatter power (Figs. 3b and 3d). In order to check the applicability of the exponential loss function we computed the depth dependence of the power contributions by means of Eq. 4, adjusted to the power

penetration depth of the multilayer model. The exponential curves are describing the average shape of the multilayer depth dependence of losses quite well. However, there is a minor shift in the top metres due to the overestimation of losses in the near-surface layer with smaller grain size.

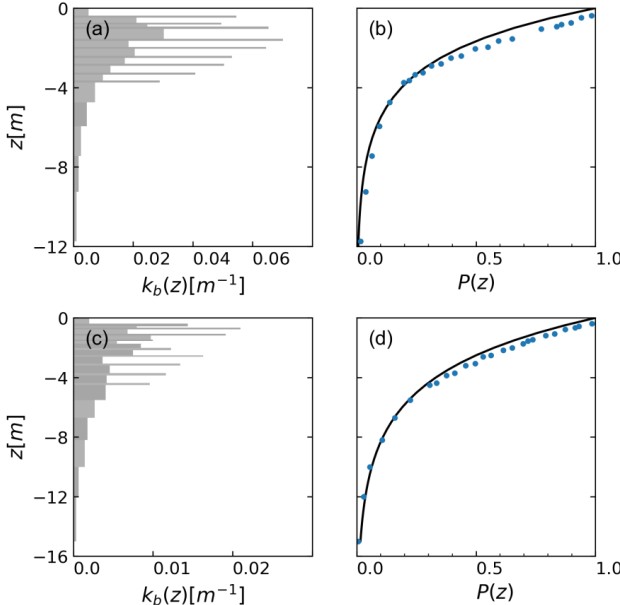

**Figure 3**. Simulations of X-band HH-polarized backscatter of a dry snow/firn volume at 40 deg. incidence angle. Input

parameters refer to Union Glacier Pit 2 (a, b) and Pit 4 (c, d). (a) and (c): Backscatter contribution of individual layers per unit volume attenuated due to propagation through the layers above. (b) and (d): Normalized contributions to total backscattered power in dependence of depth. Symbols: contributions of individual layers. Curve: Model for exponential extinction.

The DMRT simulations for $\theta_i = 40°$, using the observed vertical profiles of grain size and the estimated

stickiness, are matching the mean $\sigma°$ values for the 2013 and 2014 TDM data. However, using the same grain size and stickiness values for computing the backscatter at lower incidence angles yields an underestimation of $\sigma°$. The mean $\sigma°_{HH}$ of the TDM 2013 and 2014 data is -7.66 dB at Pit 2 and -11.96 dB at Pit 4. Using the same grain size and $\tau$ values for computing $\sigma°$ at $\theta_i = 21.8°$ (the mean $\theta_i$ of T2016 and T2018) yields -6.64 dB for Pit 2 and -10.55 dB for Pit 4, whereas the observed $\sigma°_{HH}$ values are -3.75 dB for Pit 2 and -2.85 dB for Pit 4.



Agreement between simulated and observed σ° can be achieved for the T2016/18 data by setting τ to the value 0.1 for all layers and increasing the grain size of each layer by 30% for Pit 2 and by 80 % for Pit 4, exceeding the observed size. The need for different parameter settings to compute the backscatter coefficients at different incidence angles is an indication for structural anisotropy that is not taken into account by the uniform volume approach. The increased backscatter towards vertical incidence can be attributed to increased contributions of

internal interfaces including coherent layer effects (Tan et al., 2017). The angular gradients of the backscatter coefficient show major differences between various glacier sections indicating local differences in the structural anisotropy. Pit 4 is located in a section with a high angular gradient (Fig. S2).

**4.2  InSAR coherence in a random medium**

The proposed procedure for estimating the interferometric elevation bias is based on the inversion of the

volumetric correlation factor which can be derived from total coherence products ($\gamma_{tot}$) generated during InSAR processing. The total interferometric complex correlation coefficient (coherence) of a random medium is made up by the following contributions (Krieger et al., 2007):

$$\gamma_{tot} = \gamma_{therm} \cdot \gamma_{Quant} \cdot \gamma_{Amb} \cdot \gamma_{Rg} \cdot \gamma_{Az} \cdot \gamma_{Vol} \cdot \gamma_{temp} \tag{8}$$

The terms on the right-hand side refer to the interferometric correlation coefficient related to the signal-to-noise

ratio ($\gamma_{therm}$), quantization ($\gamma_{Quant}$), azimuth and range ambiguities ($\gamma_{Amb}$), baseline decorrelation ($\gamma_{Rg}$), relative shift of the Doppler spectra ($\gamma_{Az}$), volumetric decorrelation ($\gamma_{Vol}$), and temporal decorrelation ($\gamma_{temp}$). Temporal decorrelation is not relevant for SP-InSAR data over ground, including snow and ice ($\gamma_{temp} = 1.0$).

The thermal interferometric correlation component is related to the signal-to-noise ratio (SNR) of the two SAR images by:

$$\gamma_{therm} = 1 \Big/ \sqrt{(1 + SNR_1^{-1})(1 + SNR_2^{-1})} \tag{9}$$

For SP-InSAR the volumetric correlation coefficient can be derived from the total coherence by:

$$\gamma_{Vol} = \frac{\gamma_{tot}}{\gamma_{therm}\, \gamma_{Quant}\, \gamma_{Amb}\, \gamma_{Rg}\, \gamma_{Az}} \cdot \tag{10}$$

The phase noise due to $\gamma_{Amb}$, $\gamma_{Quant}$ and $\gamma_{Az}$ of advanced SAR systems is small. For TDM SP-InSAR interferograms Krieger et al. (2007) estimate the typical loss in coherence for each of the terms $\gamma_{Amb}$, $\gamma_{Quant}$ and

$\gamma_{Az}$ at < 2%. Baseline decorrelation, $\gamma_{Rg}$, is avoided by applying common bandwidth filtering.

The interferometric vertical wavenumber, $k_z$ [rad m$^{-1}$], relates the phase of interferometric correlation to the geometric configuration of the interferometer, providing phase ($\varphi$) to height conversion:

$$k_z = \frac{\partial \varphi}{\partial z} = \frac{2\pi}{H_a} \tag{11}$$

$H_a$ is the height of ambiguity in free space:

$$H_a = \frac{\lambda\, r_0\, \sin\theta_i}{p\, B_n} \tag{12}$$



$\lambda$ is the radar wavelength, $r_0$ is the slant range distance, $\theta_i$ is the incidence angle at the air/snow interface and $B_n$ is the effective interferometric baseline. $p = 1$ is valid for the combination of one monostatic and one bi-static SAR image forming an interferogram, $p = 2$ for the combination of two monostatic images.

The wavenumber in a lossy volume accounts for the change in the propagation constant and refraction (Lei at al., 2016):

$$k_{zVol} = k_z \sqrt{\varepsilon} \, \frac{\cos \theta_i}{\cos \theta_r} = \frac{2\pi}{H_{aVol}} \qquad (13)$$

$\varepsilon$ is the dielectric permittivity, $\theta_r$ is the refraction angle and $H_{aVol}$ is the height of ambiguity in the volume. For dry snow and ice the absorption losses are very small so that the real part of the permittivity can be used (Mätzler, 1996). In Table 1 the values of $H_a$ (in vacuum) and $k_{zVol}$ are specified for the TDM scenes. We assume a snow density of 400 kg m$^{-3}$ for computing $\varepsilon$.

By applying an exponential loss function as specified in Eq. 4, Dall (2007) derived the following formulation for the complex coherence of a homogeneous infinitely deep scattering medium:

$$\gamma_{Vol} = \frac{1}{1+j2\pi d_{p2}/H_{aVol}} \qquad (14)$$

Normalizing the coherence by the interferometric phase of the volume surface, so that the coherence is 1 for zero penetration and zero when the two-way penetration depth equals $H_{aVol}/2\pi$, yields the following relation for estimating the elevation bias from the coherence phase (Dall, 2007):

$$h_b \approx \frac{\angle\gamma}{k_{zVol}} = \angle\gamma \, \frac{|H_{aVol}|}{2\pi} \qquad (15)$$

As the coherence phase in a uniform volume is uniquely defined by the coherence magnitude, the following formulation can be used for estimating the elevation bias in a uniform medium:

$$h_b \approx -\left|\frac{H_{aVol}}{2\pi}\right| \arctan\left(\sqrt{|\gamma_{Vol}|^{-2} - 1}\right) \qquad (16)$$

We apply this equation for estimating the elevation bias from the observed coherence, using the magnitude of the volumetric InSAR correlation factor as input. According to this formulation the actual InSAR elevation bias becomes progressively smaller than the two-way penetration depth with increasing relative penetration ($d_{p2}/H_{aVol}$) and approaches one quarter of $H_{aVol}$ for large ratios.

Weber Hoen and Zebker (2000) specify a formulation for the volume correlation factor in which the interferometric phase is proportional to the penetration depth:

$$\gamma_{Vol} = 1 \bigg/ \sqrt{1 + \left(\frac{p\pi \, d_l \, B_n}{r_0 \, \lambda \tan \theta_i}\right)^2} = 1 \bigg/ \sqrt{1 + \left(\frac{\pi \sqrt{\varepsilon} \, d_l \, \cos\theta_i}{H_a}\right)^2} \qquad (17)$$

This formulation is based on the same approach for the vertical scattering contributions as used for Eq. 14 and provides for small relative penetration similar results for the relation between $h_b$ and $\gamma_{Vol}$ as Eq. 16. The difference between the two approaches becomes significant if $d_{p2}/H_{aVol}$ exceeds the value of 0.1 (Fig. S3).


## 5. Analysis of the backscatter signatures, coherence and elevation bias

The analysis and discussion on backscatter signatures, coherence and elevation bias is focussing on the snow pit
sites, the BIA and the level glacier area excluding the BIA that is covered by the LGA mask the outline of which

is shown in Fig. S1 and in Figs. 8 and 9. The LGA mask covers level areas and slopes smaller than 5° inclination
and is completely covered by all of the TDM scenes listed in Table 1. The slope constraint reduces impacts of
errors in DEM co-registration and effects of different observation geometries such as radar layover and
foreshortening.

### 5.1 Spatial pattern of backscatter signals and coherence

Fig. 4 shows an image of the backscatter cross section ($\sigma°$) derived from TDM data of 6 May 2013 (scene
T2013A), 9 x 9 pixels low pass filtered. The spatial pattern of backscatter intensity on the level sections of the
main glacier and its tributaries reflects primarily differences in volume scattering properties. Low $\sigma°$ values refer
to the BIA and areas of comparatively fine grained snow/firn, whereas high values are an indication for large
scattering elements. The BIA has a comparatively smooth surface, accounting for low $\sigma°$ at an incidence angle of

40°. Low $\sigma°$ values are also evident on tributary glaciers away from the main passage of the katabatic wind and
at locations of increased accumulation rates in the vicinity of the camp. At lower incidence angles $\sigma°$ is higher
throughout and the overall dynamic range of $\sigma°$ is reduced, as evident in Fig. S1 which shows backscatter and
coherence images of 10 December 2016.

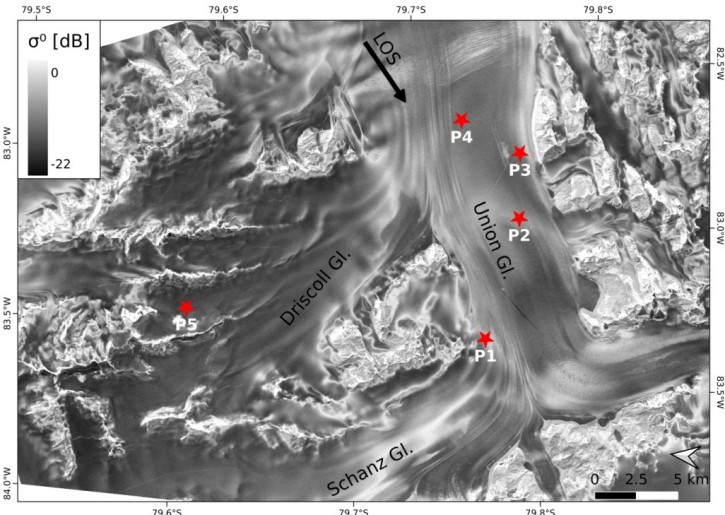

**Figure 4.** Section of TDM backscatter image, HH polarization, over Union Glacier, 6 May 2013. LOS (line of sight)
indicates the radar look direction.

The angular gradients of $\sigma°$ in the vicinity of the BIA and in crevasse zones are rather small (Fig. S2). In these
areas the differences in $\sigma°$ between the scenes T2013A ($\theta_i = 40.9°$) and T2016HH ($\theta_i = 21.6°$) amount to 2 dB to
3 dB and $\sigma°$ is high in both scenes. This is an indication for large scattering elements relative to the wavelength
such as grain clusters. Multiple scattering between individual layers and scattering at rough internal interfaces

may also play a role. In the areas with higher accumulation rates the angular differences are larger, reaching

values up to 8 dB on the area orographically left of the camp. Large angular gradients of the stratified snow/firn medium can be explained by increased backscatter contributions of internal interfaces towards near-nadir angles. The angular difference of $\sigma°$ on the BIA (-13.3 dB in scene T2013A, -5.9 dB in T2016HH) is characteristic for

backscattering of a slightly rough surface (Fung, 1994).

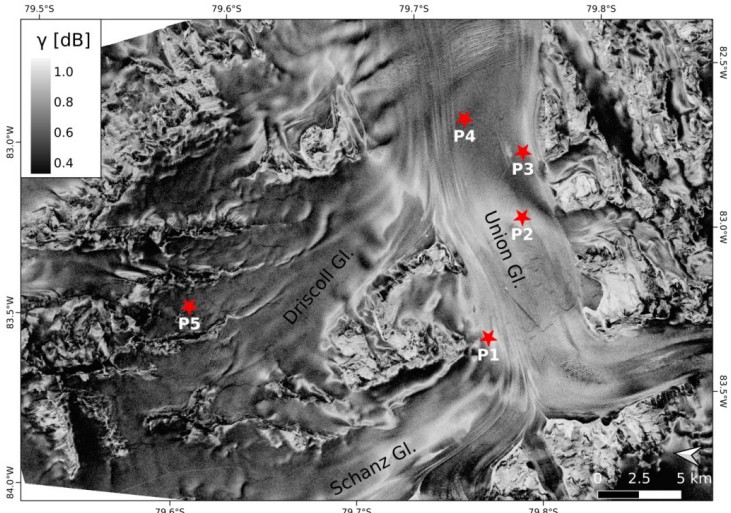

**Figure 5.** Image of total normalized coherence, $\gamma_{tot}$, from the TDM interferogram of 6 May 2013 over Union Glacier.

The coherence image of 6 May 2013 (Fig. 5) shows the lowest coherence on glacier sections with large signal penetration. In the 2013 and 2014 TDM images the coherence of the BIA (mean $\gamma_{tot} = 0.79$) is lower than in the

surrounding areas because of thermal decorrelation due to the comparatively low SNR. In the surrounding low accumulation areas the $\sigma°$ values range from -5 dB to -8 dB and the magnitude of $\gamma_{tot}$ from 0.85 to 0.90. The incidence angle has an also impact on the relation between coherence and $\sigma°$. This is evident by comparing scatterplots of two scenes with different incidence angles (Fig. S4). T2013A shows an approximately linear relation between coherence and $\sigma°$ with two cluster centres corresponding to the surroundings of the BIA and the

areas with higher accumulation rates. The T2016 scatterplot shows wide scatter with one main cluster.

### 5.2 Relation between backscatter, coherence and elevation bias

We derived the backscatter coefficient, the total and volumetric correlation coefficient and the elevation bias for cells with 90 m diameter centred at the snow pit sites (Tables S3 and S4). The azimuth resolution of the single polarization data is 3.3 m and of the dual polarized data 6.6 m. The ground range resolution is 3.34 m at $\theta_i = 21°$

and 1.86 m at $\theta_i = 40°$. This yields for the 90 m cells a speckle related uncertainty (standard deviation) of 0.13 dB for the single polarized data at $\theta_i = 40°$ (Case 1) and of 0.26 dB for the HH and VV polarized data at $\theta_i = 21°$ (Case 2). The absolute and relative radiometric accuracies for the TerraSAR-X stripmap data are estimated at 0.6 dB and 0.3 dB, respectively (Breit et al., 2010). Pit 5 is not covered by the scenes T2016 (10 December 2016) and T2018 (10 January 2018). We derived the data for Pit 5 from two scenes of adjoining tracks with similar

height of ambiguity and incidence angle: 7 January 2017 ($\theta_i = 24.6°$, $H_a = -70.0$ m) and 16 January 2018 ($\theta_i = 24.7°$, $H_a = -106.0$ m).



The coherence is computed for 11 x 11 pixels estimation windows, adding up to 390 independent samples for Case 1 and 104 independent samples for Case 2. According to the Cramer-Rao bound for coherence estimation the standard deviation for a $\gamma$-value of 0.5 is 0.03 for Case 1 and 0.05 for Case 2. These uncertainties refer to the impact of random phase noise. The uncertainty decreases towards higher $\gamma$-values (Bamler and Hartl, 1998). For the snow pit sites we use the average $\gamma$-values of the coherence pixels whose centre coordinates fit within the 90 m cell. The volumetric normalized coherence, $\gamma_{Vol}$, is derived from the observed total coherence according to Eq. 10. The thermal correlation factor is derived from the mean $\sigma°$ of the snow pit cell and the noise equivalent $\sigma°$ (NESZ). For deriving $\gamma_{Vol}$ of the 2013 and 2014 scenes the value 0.96 is used for $\gamma_{Amb} \cdot \gamma_{Quant} \cdot \gamma_{Az} \cdot \gamma_{Rg}$ and -23 dB for NESZ, the corresponding values for the 2016 and 2018 scenes are 0.97 and -24 dB.

Fig. 6 shows plots of the volumetric coherence and the backscatter coefficient versus the elevation difference dh between the TDM DEMs and the REMA at the snow pit sites. There is a clear trend of decrease in $\gamma_{Vol}$ with increasing magnitude of dh. The scene T2014A with the largest vertical wavenumber shows the smallest gradient for $dh/d\gamma_{Vol}$ and the scene T2018 with the smallest wavenumber shows the steepest gradient, as expected according to theory. The same behaviour is evident for the whole LGA data sample in which T2014A shows the lowest volumetric coherence and T2018 the highest (Table 3).

**Table 3.** Mean values over the level glacier area (LGA) for the elevation difference TDM - REMA (dh), the TDM elevation bias by inversion of volumetric coherence ($h_{bInv}$), the difference between dh and $h_{bInv}$, the volumetric coherence ($\gamma_{Vol}$) and the backscatter coefficient ($\sigma°$). $R^2$ is the coefficient of determination for linear correlation between $d_h$ and $h_{bInv}$, RMSD is the root mean square difference between $d_h$ and $h_{bInv}$.

|  | T2013A | T2013B | T2014A | T2014B | T2016H | T2016V | T2018H | T2018V |
|---|---|---|---|---|---|---|---|---|
| dh [m] | -5.97 | -5.63 | -5.49 | -5.10 | -4.28 | -4.48 | -4.78 | -4.82 |
| $h_{bInv}$ [m] | -5.80 | -5.43 | -4.85 | -4.78 | -4.39 | -4.40 | -5.17 | -5.19 |
| dh -$h_{bInv}$ [m] | -0.17 | -0.20 | -0.64 | -0.32 | 0.11 | -0.08 | 0.40 | 0.39 |
| $\gamma_{Vol}$ | 0.791 | 0.778 | 0.656 | 0.808 | 0.864 | 0.858 | 0.927 | 0.926 |
| $\sigma°$ [dB] | -9.37 | -9.95 | -8.21 | -9.12 | -5.21 | -5.36 | -4.49 | -4.71 |
| $R^2$ | 0.57 | 0.59 | 0.47 | 0.49 | 0.41 | | 0.27 | |
| RMSD [m] | 1.88 | 1.84 | 2.03 | 1.56 | 1.43 | | 1.79 | |

The incidence angle has an effect on the elevation bias as the differences between the T2013 and T2014 scenes (T2013/14) and the T2016 and T2018 scenes (T2016/18) show. For a uniform volume (Eq. 4) a larger elevation bias $|h_b|$ is expected for T2016/18 due to the steeper propagation path in the snowpack. However, the average dh value at the 5 snow pit sites is -5.34 m for (T2013/14) and -4.47 m for (T2016/18). The same trend is evident for the LGA (Table 3): dh (T2013/14) is -5.55 m, dh (T2016/18) is -4.59 m. This behaviour is an indication for non-isotropic scattering functions with increased scattering contributions towards near-nadir incidence, lifting the position of the scattering phase centre in the volume. It is also in line with the large increase of $\sigma°$ towards low incidence angles. For estimating the impact of different incidence angles we compare two scenes with (almost) the same vertical wavenumber T2013B ($\theta_i$ = 38.6°, $k_{zVol}$ = 0.121) and T2016 ($\theta_i$ = 21.6°, $k_{zVol}$ = 0.120): the dh





values are -5.63 m (T2013B) and -4.38 m (T2016). Assuming for T2016 the same uniform volume scattering and absorption properties as for T2013B the expected $h_b$ value for T2016 is -6.04 m.

Regarding polarization, there are no significant differences between HH and VV polarized data for $\gamma_{Vol}$ and dh. Whereas the snow pit sites show slightly larger dh at HH polarization, for the LGA this is the case at VV

polarization. The differences in $\sigma°$ and coherence between HH and VV polarization are also small. The average $\sigma°_{HH}$ is 0.28 dB lower than $\sigma°_{VV}$. This is in accordance with DMRT backscatter simulations showing for $\sigma°_{HH}$ values that are lower by 0.3 dB.

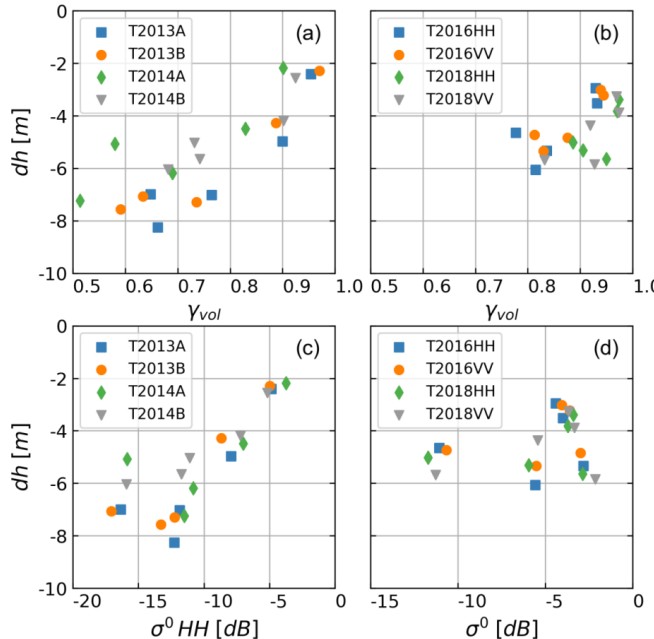

**Figure 6.** Relations between the elevation difference (dh) TDM - REMA and volumetric coherence (a, b), respectively the

backscatter coefficient (c, d) for the snow pit sites P1 to P5.

The plots of dh vs. $\sigma°$ at the snow pits in Fig. 6 indicate for the 2013/14 scenes an approximately linear relation for the sites P1 to P4, but the data of P5 ($\sigma°$ -16.3 dB) are shifted by a few dB. The reduced $\sigma°$ of P5 can be attributed to the smaller grain size and smoother vertical density profile. The T2016/18 data do not show a clear relation between dh and $\sigma°$. P4 with ($\sigma_{hh}°$ -2.8 dB) and P5 ($\sigma_{HH}°$ -11.0 dB) have a similar elevation bias. The

same behaviour as for P4, comparatively deep penetration and high $\sigma°$ in the T2016/18 data, is evident for an extended area orographically left of the camp which shows high backscatter (mean $\sigma°$ = -3 dB) and a comparatively large elevation bias (mean dh = -5m). The high $\sigma°$ at near nadir angles is an indication for increased backscatter at internal interfaces, but it is not clear why this has less impact on volume decorrelation. Reflections between individual layers and interfaces may play a role. The relation between dh and $\sigma°$ shows also

for the full LGA data set a wide spread. The coefficient of determination for linear relations between dh and $\sigma°$ on the LGA ranges from $R^2 = 0.06$ for scenes T2016 and T2018 to $R^2 = 0.21$ for scene T2013B. Part of the



spread is caused by speckle related uncertainty and uncertainty in dh, but systematic deviations in sub-regions are also evident, most likely associated with spatial variations in snow/firn microstructure and stratigraphy.

## 6.  Estimation of the interferometric elevation bias

Building on the signature analysis reported in Sect. 5, we focus on the use of the volumetric coherence for deriving the interferometric elevation bias. For inverting $\gamma_{Vol}$ in terms of dh exponential extinction in a uniform volume is assumed. The use of a multi-layer model for inversion is not feasible considering the limited dimension of the available input data. For computing the vertical wavenumber in the volume and the refraction angle we assume a snow density of 400 kg m$^{-3}$, resulting in $\varepsilon' = 1.763$ (Mätzler, 1996). Fig. 7 shows plots of the

computed elevation bias, $h_{bInv}$, at the snow pit sites derived from $\gamma_{Vol}$ vs. the elevation difference dh between the InSAR DEMs and the REMA. In order to check effects of the incidence angle the data from 2013 and 2014 and from 2016 to 2018 are displayed separately. The T2013/14 data show a highly significant linear relation between dh and $h_{bInv}$, with a coefficient of determination $R^2 = 0.86$. The root-mean-square difference (RMSD) is 0.74 m, resulting from errors of the computed $h_{bInv}$ and the DEM difference product. The T2016/18 data show a linear

relation with $R^2 = 0.59$, the RMSD is 0.84 m.

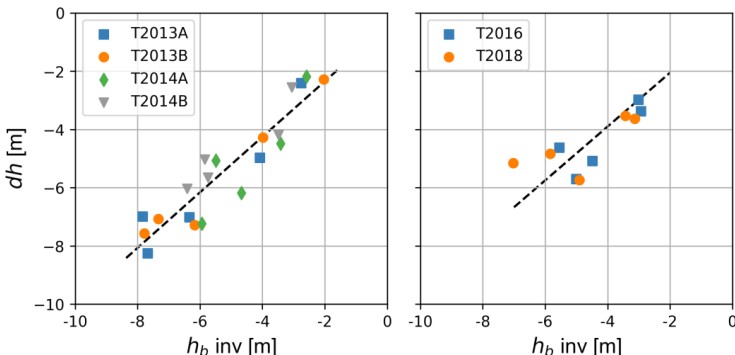

**Figure 7**. Elevation difference (dh) TDM DEM - REMA versus the computed elevation bias $h_{bInv}$ derived from the volumetric coherence for the snow pit sites P1 to P5. - - - linear regression line.

Maps of dh and the computed TDM elevation bias are shown in Fig. 8 for scene T2013B and in Fig. 9 for scene

T2016. These two scenes have almost the same vertical wavenumber but different incidence angles. The differences between HH and VV polarized data of the T2016 and T2018 scenes are insignificant. We use the mean value of the HH and VV based DEMs of the single dates for the statistical comparison in order to reduce the impact of random phase noise. In Table 3 mean numbers over the LGA are specified for dh, $h_{bInv}$ and $\gamma_{Vol}$, the coefficient of determination ($R^2$) for linear correlation between dh and $h_{bInv}$ and the root-mean square difference

(RMSD). The numbers for $R^2$ and RMSD refer to the maps resampled to 8 m grid size, low-pass filtered over 7 x 7 pixels windows using a Gauss function. The dh value of the TDMgl DEM (dh = 5.61 m) differs only by 0.06 m from the mean dh of the T2013/14 data. It is based on several TDM scenes acquired in 2013 and 2014. The dh map for TDMgl vs. REMA shows a similar spatial pattern as dh of the individual DEMs (Fig. S5).



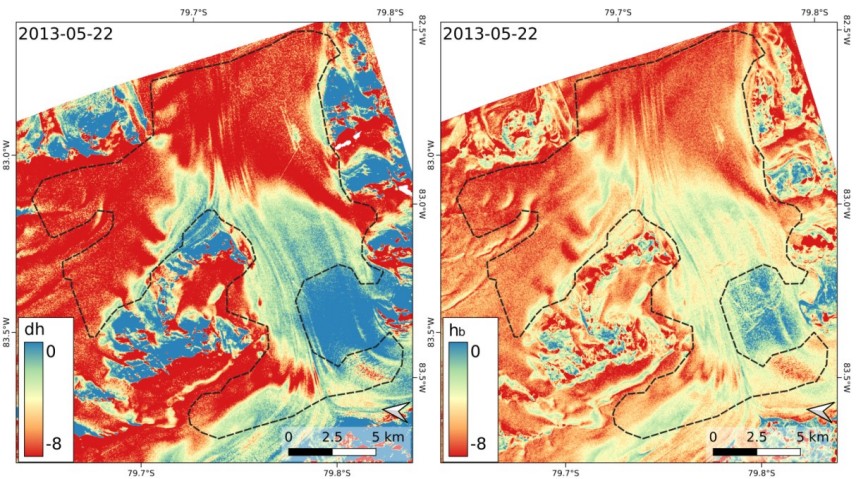


**Figure 8.** Elevation difference (dh) TDM DEM – REMA and elevation bias ($h_b$) by inversion of $\gamma_{Vol}$ for the TDM scene 22 May 2013. The outline encloses the LGA.

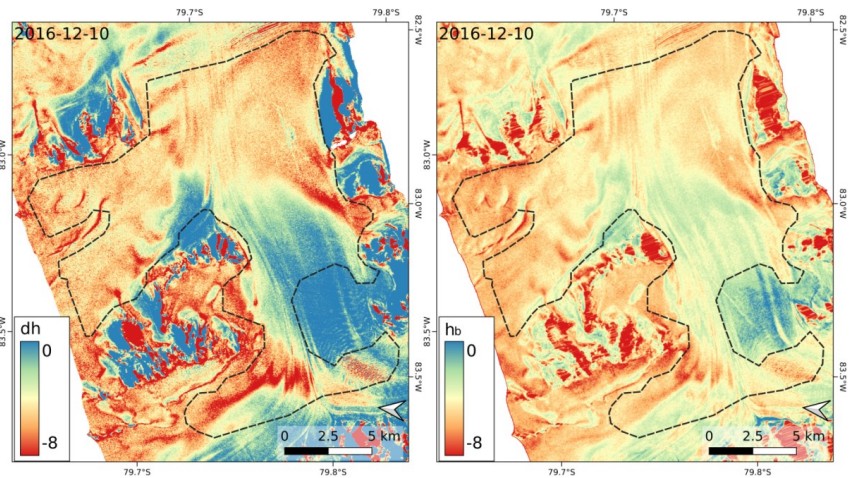

**Figure 9.** Elevation difference (dh) TDM DEM – REMA and elevation bias ($h_b$) by inversion of $\gamma_{Vol}$ for the TDM scene 10
December 2016, based on HH and VV polarized data.

As for the snow pit sites, the mean values over the LGA show distinct differences in dh, $h_{bInv}$ and $\gamma_{Vol}$ between
the data sets with different incidence angle. The magnitudes of the elevation bias of the T2013/14 data (mean dh
= -5.55 m, $h_{bInv}$ = -5.22 m) are larger than the corresponding values of the T2016/18 data set (dh = -4.59 m, $h_{bInv}$
= -4.79 m). As for the snow pit sites, this is opposite to the expectation for a uniform isotropic scattering medium
for which |$h_b$| should be larger for smaller off-nadir angles. The scenes with similar vertical wavenumber show
dh = -5.63 m for T2013B and dh = -4.38 m for T2016 whereas the expected value for T2016 is -6.19 m if the
same absorption and scattering properties are assumed as for T2013B.

The LGA mean values of dh and $h_{bInv}$ show minor differences: -0.32 m for T2013/14 and 0.20 m for T2016/18.
The spatial patterns of dh and $h_{bInv}$ are similar, but the mean slope of the 2D distribution deviates from the 1:1



correspondence (Fig. S6). The magnitude of the computed elevation bias is overestimated over the areas with coarse grained firn and small penetration depth in the surroundings of the BIA and underestimated in areas of higher accumulation rate. These depth dependent deviations can, at least partly, be attributed to the simplified assumption of the uniform volume approach. In the low accumulation areas the large grains, grain clusters and rough interfaces in the top 2 m are efficient scattering sources whereas the high density below causes reduced

scattering due to the dense medium effect. In the high accumulation areas the depth of the scattering centre is underestimated because of reduced scattering in surface layers with smaller grains.

### 7. Discussion and conclusion

Single-pass across-track SAR interferometry is an efficient technique for comprehensive, spatially detailed measurements of glacier and ice sheet topography as the measurements are not impaired by temporal

decorrelation of the interferometric phase, variations in atmospheric propagation conditions, cloudiness, and variable illumination. However, the penetration of the InSAR signal into snow and ice introduces a bias in the measured elevation. Empirical and model-based approaches have been applied for correcting the elevation bias in InSAR DEMs. A common approach is the use of laser altimetry data as reference for vertical co-registration. However, the altimetry data often lack the required temporal coincidence and spatial coverage. For repeat

measurements with the same SP-InSAR sensor and observation geometry the backscatter coefficients and coherence can be used as indicators for stability or changes of snow volume properties and signal penetration. In case of unchanged backscatter and coherence values no correction is required for deriving glacier elevation change (Rott et al., 2018). However, for the majority of applications corrections for penetration are needed, in particular if elevation data from different sensors are used.

The theoretical and experimental work reported by Weber Hoen and Zebker (2000) and Dall (2007) suggests the use of the volumetric coherence as a key parameter for estimating the InSAR signal penetration. In order to evaluate the suitability of this approach for polar snow and firn we analysed multiple SP-InSAR data of the TanDEM-X mission over Union Glacier. The inversion of a single parameter does not allow the detailed representation of the complex layered structure of natural snow. Therefore we use a simplified model, the

uniform volume approach, for describing the scattering and absorption properties. In order to check the suitability of this approach for inferring the InSAR signal propagation in snow and firn volumes with different structural properties we performed simulations with a multi-layer dense medium radiative transfer model. The computations show large variations of the backscatter coefficient between individual layers but the average depth-dependent backscatter contributions can be approximated by an exponential function that reflects the

extinction in a uniform medium. However, the model underestimates the increase of backscatter towards low incidence angles. This can be explained by structural anisotropy in which horizontally oriented scattering elements are causing increased backscatter towards near-nadir angles (Nghiem et al., 1993). Measurements of fabric and microstructure in the top 3 m of snow on the East Antarctic plateau show minor structural anisotropy for grain layers that are subject to strong temperature gradients (Calonne et al., 2017). Randomly orientated c-

axes are observed in denser layers of rounded grains, whereas a preferential orientation in the horizontal plane shows up in large faceted grain layers. However, such layers are absent in the wind exposed zones of Union Glacier and marginal in other sections, excluding anisotropy of grains as a main explanation for the observed angular backscatter behaviour.

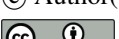



A most likely explanation is increased forward scattering at internal interfaces and wind-packed structures. In particular on the main section of Union Glacier the snow surfaces are wind roughened, showed elongated sastrugi of metre length with steps and ripples of several centimetres vertical extent. The surface roughness related to wind packing and erosion is also evident in snow pits at interfaces between snow layers and wind crusts. Ashcraft and Long (2006) attribute the backscatter anisotropy, observed in scatterometer data of katabatic wind zones, to the anisotropy in the preferential roughness direction of the snow surface and internal boundaries. Such behaviour was also observed for density stratified firn on the East Antarctic Plateau and reproduced by simulations with a layered-medium RT model (Rott et al., 1993; West et al., 1996).

The observed differences of the elevation bias at different incidence angles are another evidence for structural anisotropy. For a uniform lossy medium larger values of the penetration depth and elevation bias are expected at near nadir angles. However, the 2013/2014 TDM data and 2016/2018 TDM data show the opposite behaviour. As for the observed angular backscatter gradients this can be explained by increased backscatter contributions at internal interfaces at small off-nadir angles.

Constraints of the uniform volume model are also evident in the deviation from an exact 1:1 correspondence between the observed elevation difference dh and the computed elevation bias, $h_{bInv}$. Whereas the mean values of dh and $h_{bInv}$ show only minor differences, a trend towards overestimation of the computed penetration is evident for the heavily wind-exposed areas and towards underestimation for areas with reduced impact of wind and deeper signal penetration. In the first case large grains, clusters and wind crusts in the top layers act as strong scattering elements whereas the high density causes a pronounced decrease of scattering with depth. Multi-layer interactions between the internal interfaces may also play a role as observed in tomographic experiments (Fischer et al., 2019a). In the areas with higher accumulation rates the top snow layers show below-average grain size implying reduced scattering. In this case the assumption of uniform extinction properties introduces an underestimation of the computed elevation bias. This effect was also observed in the percolation zone of Greenland by Fischer at al. (2019b) where the assumption of uniform scattering and extinction leads to an underestimation of the phase centre depth due to below-average grain size in the top layer.

In spite of the issues addressed above, the average values of the elevation bias over Union Glacier, derived from volumetric coherence, agree well with the mean elevation difference between the TDM and optical DEMs. The spatial patterns of the simulated and observed elevation bias are also quite similar. As the deviations from the 1:1 correspondence of dh and $h_{bInv}$ are not random but related to spatial variations of the snow and firn structure, advancements for the estimation of the InSAR elevation bias can be expected from progress in the representation of snow/firn structural properties in models for radar signal propagation. After all, the derivation of the interferometric elevation bias from volumetric coherence is a promising option which should be carried forward as it delivers spatially detailed information coinciding both in space and time with the topographic product.

Fischer et al. (2020) observed also deviations from the uniform volume approach for deriving the volumetric coherence from airborne multi-frequency polarimetric SAR data. They explored sources and interaction mechanisms responsible for these deviations and tested different models for vertical backscatter contributions, concluding that in case of single polarization data the inversion based on the uniform volume model is a preferred approach. Progress for locating the scattering sources in layered polar firn and estimating the InSAR penetration bias can be expected from the use of polarimetric radar data. Parrella et al. (2016) developed a polarimetric scattering model accounting for structural anisotropy in layered firn. They applied this model to



polarimetric L-band SAR data for deriving the shape and orientation of scattering elements in the firn volume. Fischer at al. (2019a) derived the depth of dominant scattering layers in firn from multi-baseline polarimetric interferometric SAR data in different frequency bands, including X-band. Consequently, further progress on InSAR signal penetration in layered media can be expected from the use of polarimetric data as well as from multi-baseline and multi-angle approaches.

*Data availability*. Interferometric satellite products processed for this study will be made available upon publication of the final version of the manuscript at http://cryoportal.enveo.at/.

*Supplement.* The supplement related to this article is available on-line.

*Author contributions*. HR conceived the study, performed the field work, was in charge of the data analysis and scientific interpretation and drafted the manuscript. SS and JW contributed to numerical backscatter simulations
and the analysis of topographic data. LK processed and calibrated the interferometric satellite data. All authors contributed to the data interpretation, discussion of the results and revision of the manuscript.

*Competing interests*. The authors declare that they have no conflict of interest.

*Acknowledgements*. The TanDEM-X data were made available by DLR through the projects XTI_GLAC6809 and DEM_GLA1059. The ICESat and ICESat-2 laser altimeter data were obtained from the NASA Distributed
Active Archive Center, US National Snow and Ice Data Center (NSIDC), Boulder, Colorado. REMA data were downloaded from the U.S. Polar Geospatial Center. Landsat 8 images were downloaded from the United States Geological Survey (USGS) Landsat Archive. HR would like to thank Antarctic Logistics & Expeditions LLC (ALE) for perfect logistic support and for the provision of meteorological data and in particular Nate Opp for his active and knowledgeable support in the field work.

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
