# Peer review of "Penetration of interferometric radar signals in Antarctic snow"

_The Cryosphere, 2020_

## Author Response (AR1)

**Manuscript tc-2020-380, Author's response, AR1**

**General Response**

We are grateful to the referees for the time and efforts they have taken to provide such detailed and valuable comments. The comments from both referees are well-founded. We implemented the suggested changes except in a few cases were these seem to be based on misunderstanding regarding technical issues. In those cases we provide now better explanations on the related physical and technical background. We are confident that the changes improve the manuscript significantly, providing a better focus on the main issues and improving the readability.

The changes are outlined in detail in the response to the referees. Main changes include:

- In several sections the text was revised and shortened in order to facilitate the understanding and provide a better focus.
- We omitted some of the technical details. The revised manuscript contains the technical information that is essential for traceability and understanding.
- Where appropriate, we added short explanations on the motivation for performing specific tasks.
- We re-arranged the sequence of some material and text as follows:

- Information on the uncertainty of backscatter and coherence maps was shifted from Sect. 4 to Sect. 2.1 (TanDEM-X data).
- We shortened the methodological information on interferometric coherence and signal penetration (previously in Sect. 3.2) and show it in the revised Sect 3 which deals now only with this topic.
- The background information on the vertical backscatter profile and the study on backscatter simulations for the snow pit sites (previously in Sect. 3.1) were shifted to the Appendix, with some shortening.
- A separate section (Sect. 6) deals now with the discussion (including some of the material in the previous Sect 4, 6 and 7)
- Sect. 7 includes now only the discussion.

The changes are highlighted in the attached tracked-changes document. A main part of the highlighted changes is related to the shifting of text passages or subsections of the initial manuscript version to other sections, rather than presenting completely new information.

*Referee comments are in italics,* our responses and explanations on implemented changes are in normal font.

The line numbers in the response refer to the revised version of the manuscript.

**Response to Referee 1**

We wish to thank the Referee for the valuable comments and suggestions which are very helpful for improving the manuscript. We address the comments below and explain how these were taken into account in the revised manuscript.

*General Comment*: *This paper concerns a new application of existing theory to estimate bias in InSAR-derived elevation measurements of ice sheets and glaciers caused by below-surface radar returns. The method is based on the relationship with volume scattering coherence, which can be determined from total coherence, knowledge of the signal to noise ratio and assumptions of coherence loss from smaller order terms. The estimated bias from volume scattering coherence measurement is*

*compared with bias calculated from the difference between InSAR elevation measurements and REMA.*

*There is no question as to the importance of this study. Observation error of cryosphere changes from InSAR is essential, particularly given the high magnitudes observed (mean bias of around 5m) and potential for seasonal variation due to changes in the snow grain size and density. The consistency between the simulation results and observed bias is encouraging. This is a new, practical application of the theoretical work of Dall 2007.*

Response: We appreciate the positive feedback on the scope of our work. This comment very well reflects the main objectives of the paper.

*Main Comment 1*: *My main concern about this study is that the Dall 2007 model is not applicable where there is significant surface scattering. While this may be valid considering the snow surface, there are a number of ice lenses and crusts within the snowpack (Figure 2) that act in a similar way. It was not clear from the snow backscatter modeling (section 4.1) how these were simulated but possible these were explicitly taken into account (e.g. ice layer with 2mm air bubbles). Discussion of the limitations of the Dall 2007 model should be included as well as the limitations of the methodology of the snow backscattering modeling. The impact of the use a subjective grain size and assumed stickiness on the retrieved bias should be discussed.*

Response: The scattering contribution of dry snow and firn surfaces at X-band and low radar frequencies is very small because of the low dielectric contrast, in particular over the glaciers where the signal of the (semi-infinite) volume below is larger by more than one order of magnitude. The scattering at internal interfaces within the volume is implicitly taken into account in the vertical loss function of the Dall model. The inversion of the Dall model is not based on the multi-layer radiative transfer model that we use for backscatter simulations. The inversion of such a model is not possible with single channel SAR data. The main objective of the reported backscatter forward modelling activity is to assess the suitability of the exponential loss function for describing the vertical backscatter contributions of a layered polar snow/firn medium.

The information on the assumptions for the multilayer backscatter model, the modelling result, and the constraints of this model, as well as of the exponential loss function, have been shifted to the Appendix and are now presented there with better focus. Limitations are discussed in the Appendix and in Sect. 6 (Discussion). The comparison of the elevation bias computed with the inverted Dall model with the penetration estimated with optical reference data shows on the average good performance, confirming the general applicability of the Dall model for dry polar firn. However, there is some over-, respectively under-estimation of the retrieved elevation bias for sites with specific structural properties.

*Main Comment 2*: *Overall the paper is detailed and would probably be better suited to a remote sensing journal in its current form. For broader applicability as a publication in The Cryosphere it needs to be more succinct. I would encourage the authors to look again at the balance of what must be provided for reproducibility, what is required for basic understanding and what information may be already available for those who really need to know the detail. For example, equations 11-15 may be better kept in the Dall 2007 reference as the jump from equation 10 to 16 will be easier to read for most. This may also allow some of the figures in the supplementary material to be brought into the main paper.*

Response: Many thanks for pointing this out. Taking into account this comment, we revised and shortened the text accordingly, streamlined the technical information and rearranged the sequence of the presented material (see the General response).

Regarding our motivation for submitting the paper to *The Cryosphere* we want to point out that many papers using TanDEM-X data for measuring glacier surface elevation change were published in this journal and the correction of the penetration related elevation bias was repeatedly addressed as a critical issue.

*Main Comment 3*: *There is a lot of detail early in the paper on ICESat / ICESat-2 (section 2.2 – nearly a page) yet these are not actually used for the estimation of InSAR elevation bias, despite them being the observation with the lowest error. There is an indication in Table S1 for the ice-free slope and blue ice area, and for the area around pit P4 in Table S2 but not over the larger area. In table S2 the standard deviation is much higher than the actual measurement. This, and the positive height biases shown in Figure 1 should be discussed – what does it mean when the TDM DEM elevation is higher than the optical-derived DEM elevation?*

Response: Regarding these issues, we want at first point out that the different geodetic data refer to different absolute (or relative) references and/or different handling regarding the interferometric penetration bias. Relevant information on the various topographic products from different sensors is provided in Sect. 2.1 and Sect. 2.2. The notations for the elevation differences between optical and SAR data are defined in Sect. 4.1.1 (Sect. 3 in the original manuscript). These include: the penetration-related InSAR elevation bias (Eq. 12), the height difference between non co-registered DEMs (Eq. 13) and the height difference between optical and InSAR elevation data, co-registered at surface scattering targets (Eq. 14). The discrimination of these parameters provides traceability of the key processing steps reported in the paper. For the TanDEM-X global DEM penetration corrections are applied over Antarctica (explained in Sect. 2.1). Because it is an operational product, available to the scientific community, we don't want to modify it. This explains the negative values for the elevation differences (ICESat minus TDM global DEM) in Fig. 1, referring to areas where the actual penetration bias is smaller than the bulk penetration correction.

A detailed account on the temporal stability of surface elevation and vertical co-registration is essential for DEM differencing and evaluation of the retrieved elevation bias, key topics of the work presented in the paper. In the revised manuscript these issues are covered in Sect 4.1.2 and 4.1.3, taken over from Sect 3.1 and 3.2 of the original manuscript after some minor revisions in order to improve the readability. For Sect. 2.2 we did some shortening, but maintain key information on the properties of the optical satellite data products used in the study.

*Main Comment 4*: *Please could the authors check for consistency throughout the paper. For example, TDM is defined as the TanDEM-X mission, then TanDEM-X is used interchangeably with TDM in lines 60-70. DEM is defined twice. There are two separate definitions of Δh and dh, with opposite signs. SMRT is suggested as the backscatter model used, but then the rest of the text refers to DMRT. If DMRT then the version used needs to be stated. Line 553 refers to equation 4, but this is not the correct equation. These are all minor defects, but unfortunately make the paper difficult to follow.*

Response: Many thanks for pointing this out. We thoroughly checked the manuscript for inconsistencies and implemented corrections where relevant. Regarding the term TanDEM-X we want to point out that this term is actually used for different items, the TanDEM-X mission, the TanDEM-X satellite itself, as part of the proper name of the TanDEM-X global DEM (c), and as part of the proper name for the operational TanDEM-X processor (ITP). We replaced the term TanDEM-X by TDM where appropriate. Regarding Δh and dh, these are two different quantities referring to different steps (intermediate products) in the procedure for deriving the elevation bias. We want to maintain these in order to provide full traceability (see response to Main Comment 3). We omit the term DMRT, except in the context of one specific scattering model (DMRT-QMS) in the Appendix where we mention also the full name.

The reference to Eq. 4 (describing the vertical backscatter function) in line 553 of the original manuscript version is correct. In the revised manuscript we shifted the description of the backscatter function and backscatter modelling to the Appendix and revised the related text.

*Specific comments (SC):*

*SC1: The abstract attributes angular gradients of backscatter intensity to anisotropy in the snow structure. This is misleading. Even if the summer observations at one angle can be compared to the winter observations at another angle (and I'm not convinced they can), this is a stratigraphic effect rather than anisotropy.*

Response: We replace the comment on the angular backscatter dependence in the abstract (not a main issue) with a sentence on the impact of internal layers on deviations from the uniform volume approach (L28 to L31). Seasonal effects on backscatter intensity in the Antarctic dry snow zone are very modest, because the observed signal is made up by volume backscatter contributions down to several metres depth and seasonal differences would affect only a thin surface layer (if at all). We ordered and analysed TanDEM-X data from July 2017 from the same orbit as the T2016 (Dec.2016) and 2018 (Jan. 2018) scenes. The backscatter intensities (sigma-0) in July are between the sigma-0 values of Dec. 2016 and Jan. 2018.

*SC2: The Section 2.1. Product reference should already contain the majority of this detail. Only additional processing steps done for this study need to be included.*

Response: TanDEM-X has about 100 different operation and acquisition modes. Therefore it is important to provide specifications for the products used in this study.

*SC3: Line 134 – please show the location of the 11 blocks (or was this part of a different study?)*

Response: The blocks used for correcting the penetration during the production of the TanDEM-X global DEM are distributed all over Antarctica (at large distances). Details and a related figure are shown in Rizzoli et al., 2017. Explained in L132 to L135 of the revised manuscript.

*SC4: Figure 2. Snow grain size legend is different in colour to the main plots. Please could you increase the snow grain type font size and/or resolution.*

Response: We changed the colour of the legend and increased the snow grain type font size.

*SC5: Section 3.2 Perhaps the processing steps would be better placed in the supplementary material. There is a lot of detail on the accuracy of REMA. It would be better to state the vertically registered DEM is treated as the truth, the errors briefly discussed as a limitation of the study and the reader referred to the supplement for more information.*

Response: The vertical co-registration and potential errors is very important for deriving height differences between different topographic data sets. Therefore we prefer retaining this section in the main paper. We slightly shortened the text (Set. 4.1.3 in the revised manuscript)

*SC6: Line 377. Stickiness of 0 breaks theoretical limits. The minimum stickiness is bounded by equation 35 in Löwe and Picard (2015).*

Response: Wee removed this statement. According to the theory of electromagnetic wave propagation in dense random media stickiness zero corresponds to infinite stickiness (Tsang et al., 2013).

*SC7: Line 517. The two observations were taken 2.5 years apart. What microstructural changes could reasonably be expected during this time period, and what would the impact be on the backscatter / elevation bias estimate? The difference has been attributed to incidence angle, but other factors have not been discussed.*

Response: The backscatter signatures, as well as the surface elevation, are remarkably stable over years (typical for the dry snow zone of Antarctica). This was one of the reasons for selecting this area for the penetration study. See also the response to SC1 referring to the lack of seasonal variation. The mean differences in sigma-0 between the 2016 and 2018 data on the snow pit sites (Table S3) and the main glacier area (Table 3) are within the absolute radiometric uncertainty for the difference (0.85 dB). Some additional (minor) uncertainty in sigma-0 can be attributed to geocoding because the six scenes used for the study were acquired from five different orbits. The elevation bias estimate is based on coherence. The backscatter intensity is only used for computing the signal-to-noise ratio which is a minor factor for deriving the volumetric coherence from the total coherence.

Significant deviations of the angular dependence of sigma-0 from isotropic scattering are typical for density-stratified polar firn. Ground-based scatterometer measurements in the dry snow zone of Dronning Maud Land, East Antarctica, show for X-band co-polarized sigma-0 differences of 5 dB to 6 dB between 20 and 40 degree incidence angles (Rott et al., 1993), similar to sigma-0 of the Pit 3 and Pit 5 sites on Union Glacier. See also L708 to L712 in the Appendix.

*SC8: Line 590 hbinv is mentioned but not defined – presumably this is from rearrangement of equation 16? It is not clear why equation 17 been included in this paper – I think this is used to calculate the volume coherence from the exponential fit to the SMRT / DMRT backscatter curves for retrieval bias but it would help the reader to state clearly the steps taken.*

Response: Thanks for pointing this out. $h_{bInv}$ is now defined in Sect. 4.1.1, L333. In Sect. 5, L482, we refer explicitly to the equation that is used for inverting the volumetric coherence in terms of the interferometric elevation bias (Eq. 11 in the revised manuscript).

**Manuscript tc-2020-380, Author comment AC2**

**Author Response to Referee 2**

We wish to thank the Referee for the valuable and detailed comments and suggestions which are very helpful for improving the manuscript. We address the comments below and explain how these are taken into account in the revised manuscript.

*General Comments:*

*In the manuscript "Penetration of interferometric radar signals in Antarctic snow" the authors study the relation between radar penetration depth into Antarctic snow and the interferometric coherence obtained from the single-pass InSAR mission TanDEM-X. They apply great effort for vertical alignment of different elevation models. For inversion they apply a model developed by (Dall, 2007) which assumes an uniform scattering efficiency of the snow volume. Based on snow pit measurements and depth-resolved radar backscatter models they conclude that despite a strong vertical variability of the scattering efficiency the depth-integrated backscatter signal represents well the model suggested by (Dall, 2007). They also found that modeling of the backscatter signal with the SMRT model based on grain size and layer thickness cannot explain the strong incidence angle dependence of the observed backscatter signal.*

*These findings make the manuscript in general suitable for publication in The Cryosphere. However, I have major concerns about the objectiveness of presented results and think that the manuscript requires a major restructuring to present and to focus on the most relevant results listed above. Below I first detail my main concerns on section 4-6, followed by minor comments and technical corrections. I would also suggest the authors to use the common structure of Data - Method - Results - Discussion*

*- Conclusion and to write the manuscript more concise, instead of the currently used sequential form of sub-method-sub-results mixed with interpretation and discussion.*

Response: Many thanks for these suggestions. We performed major restructuring and revision of the text, providing a better focus on the main issues and improving the readability. See the General Response. The object of the modelling activity is not a main issue, but aimed at checking the vertical backscatter distribution of the uniform volume model by means of simulations with a multilayer backscatter models. Therefore we shifted the related report and results to the Appendix and revised the text in order to deliver a more concise message. The discussion issues are now addressed in Sect. 6 (Discussion) and were taken out from other sections.

***Major comments:***

*Section 4.1: This section about modeling the backscatter contributions from different layers lacks a thoroughly analysis and results seem to be presented in a selective manner. I understand that modeling the backscatter signal from the complex snow structure is challenging. Therefore, I think results should be presented in a more objective way to allow the reader to draw his own conclusions. Specifically, I have the following comments:*

Response: As explained above, the backscatter modelling activities are not a main objective of the paper but are serving the purpose to check the validity of the vertical backscatter function of the uniform volume approach. We revised and shortened the account on the vertical backscatter function in order to provide a better focus on the objective and moved it to the Appendix. For in depth studies a backscatter model accounting for the complex layered structures of polar firn is needed, including i.a. the representation of coherent and incoherent scattering at internal rough interfaces and interferences between individual layers. We address this issue in the Appendix, L705 to L714, and also in Sect. 6 (Discussion).

*- line 268: Why did the author choose 25 layers and not the 30+ layers shown for the snow pits?*

Response: For the revised paper we performed simulations accounting for a larger number of layers (ca. 50). The resulting vertical backscatter functions are shown in Fig. A1. These functions are in close agreement with the computations shown in the original version of the paper. The variations between individual layers are implicitly smoothed out in the vertical backscatter profile.

*- line 370: When adopting the density profiles from firn cores, how was the layer thickness chosen? Fig. 3 indicates that, despite using the same firn core, different layer thicknesses were used below 2 meters. Please also mention that half of the simulated backscatter contribution originates from the adopted firn core. Only the upper half of simulated backscatter is based on now pit data.*

Response: Down to the snow pit base we merged layers with similar scattering properties (density, grain size). Below the snow pit depth we account for two layers per year (summer, other seasons), adapting the annual layer thickness to match the vertical density profile and taking into account the estimated annual accumulation rate. The change of grain size with depth is based on the model of Linow et al. (2012) (Appendix, L664 to L667). The backscatter profiles are shown in Fig. A1.

*Fig. 3: What are the "spikes" in Fig. 3(a) and (c)?*

Response: These are normalized backscatter contributions [1/m] of layers with high backscatter coefficients featuring larger effective grain size and stickiness than adjoining layers. For the revised paper we decided not to show the scattering contributions of individual layers. For in depth analysis a model is needed that accounts also for multiple interactions between scattering in the volume and at internal interfaces.

*line 388: "Fig. 3 shows ... simulations for 40° ... of Pit 2 and 4." Why were these two pits chosen for the figure? Why not showing simulations for all snow pits (at least for one incidence angle)?*

Response: In the revised version we show the backscatter functions for Pit 2, 3, 4, and 5 (Fig. A1). The main conclusion regarding the use of an exponential backscatter function is the same. The model is not able to reproduce the observed backscatter intensities at 20°, and for Pit 1 also at 40°. This issue is explained in the Appendix, L703 to L714.

*- line 393: "Both values differ less than 0.3 dB from the mean values of the 2013 and 2014 TDM scenes (40 deg)": This information is meaningless, considering that the standard deviation of the 2013 and 2014 measurements is around 1 dB; further, as stated by the authors, for 22 degree incidence angle, the model deviates 3 to 8 dB from the measurements and requires more tuning for a reasonable agreement.*

Response: We omit this statement.

*line 395-397: The modeled two-way penetration depths for Pit 2 is 4.72m and 7.25m for Pit 4. The distance between the snow pits is about 7 km, but modeled results differ by 2.5 meters apparently due to different snow properties. Therefore I think it is meaningless to compare the penetration depth of Pit 2 with similar results from East Antarctica (Rott 1993) except for increasing the self-citation index. Please remove the reference or provide a more tracable comparision.*

Response: This paper shows active and passive X- and C-band microwave signatures (at incidence angles between 10 and 60 deg.) measured in Dronning Maud Land at several sites with dry polar firn of different structural properties and accumulation rates. We are not aware of comparable data sets. The backscatter signatures on Union Glacier are similar to those measured in Dronning Maud Land which are characteristic for large areas of the Antarctic dry snow zone. This points out that the study of Union Glacier is of relevance for extended areas of the Antarctic dry snow zone rather than examining only a local anomaly. We explain this in L708 to L711.

*To present more objective results, I suggest to show a scatter plot presenting modeled vs. simulated backscatter intensity for all(!) 30 or 40 backscatter values listed in Table S3. Then the authors can discuss in a more objective manner what they think what causes the strong discrepancies. Different symbols or color could allow to separate different incidence angles and test sites(e.g., you could use numbers as plot symbols referring to the snow pits).*

Response: Regarding the suggestion to perform backscatter simulations separately for each of the 40 sigma-0 values in Table S3 we want to point out the following: (i) there are only five input data sets available for backscatter simulations (based on snow parameters at the five snow pit sites); (ii) there are two incidence angle classes (22 deg. and ~40 deg.) with little angular variability within each class; (iii) The differences in sigma-0 within each of the two angular classes are close to the radiometric uncertainty. Therefore we performed backscatter simulations for the five snow pit sites and two incidence angle classes. Out of these, 4 cases (shown in Fig. A1) are able to reproduce the observed backscatter intensity. All this is explained in the Appendix. For the original version of the paper we selected two typical examples. The results of the simulations for the 4 cases shown in Fig. A1 lead to the same conclusions.

*Section 5.2 is extremely hard to read. The main message of this section is not clear and backscatter results and discussions are mixed with inversion results of the penetration depth. Results from the snow pit locations are mixed with area-wide maps and scatter plots representing the same variables. Please restructure this section thoroughly and present only the most relevant results (estimation of*

*interferometric bias) in a well-structured way. Currently, it's not clear to me why you also discuss and interpret backscatter values in this section. Specific suggestions:*

Response: We shifted the part on the incidence angle dependence of the elevation bias to Sect. 6 (Discussion) and revised the text. We shifted the information on product resolution (line 523 to 525 of the original paper) to Sect. 2.1 (TanDEM-X data).

*- line 522-540 should go to the methods.*

Response: See comment above.

*- Figures and tables: Why not showing a full page or full column figure with six rows and two or three colums of subplots (6 TDM scenes x 3 types or scatter plot) where each subplot shows a scatter plot of 1:(gamma_vol over dh), 2:(h_binv over dh) and possibly 3:(sigma_0 over dh). For each subplot, you could indicate the datapoint corresponding to a snow pit with a black symbol. This would then provide a solid basis for discussion of the inversion results with different incidence angles and baselines. At the same time you avoid discussion of mean values calculated over the strongly inhomogeneous areas of the LGA.*

Response: We focused on typical examples for pointing out the main differences in order to avoid excessive length. There is always a trade-off because most readers would not be interested in minor details. We added scatterplots of additional cases to Fig. S2 and S5. They show the same features as the examples shown in the original version of the figures, in line with the conclusions reported in the original paper.

*- I think you could merge Section 5.2 with 5.3*

Response: There was no section 5.3 in the original version of the paper (and there is none in the revised version).

*- following a result-section about penetration estimation, you could - if it's worth - add a section presenting result on backscatter.*

Response: We prefer showing results on penetration and its relation to coherence and backscatter jointly, in order to highlight communalities and differences (Sect. 4.3 in the revised paper).

***Minor comments:***

*abstract, l.23. "The average depth-dependent... can be approximated.." It's not clear if that's a general statement or a finding of the study. Please indicate.*

Response: We clarified this issue (Abstract, L23 to L25)

*abstract l.29: "The angular gradients of the backscatter intensity": Unclear what "angular gradients" are. Looking at line 420-427 I think you mean that simulated backscatter data do not match the backscatter measurements at different incidence-angles?*

Response: We replaced the statement on incidence angle dependence of backscatter with information on the performance of the computed elevation bias (which is of more importance) (L28 to L33)

*line 42-44: "Backscatter contributions ... within a volume scattering medium, observed under slightly different incidence angles, are causing a spectral wavenumber shift and decorrelation (Gatelli 1994)": I think that two different things have been mixed in this sentence. The observation under slightly different incidence angles (or InSAR nbaselines) causes a phase ramp (flat earth phase) modulated by topography (topographic phase). The sum of these two phases causes the spectral wavenumber shift which can be corrected for by spectral filtering (Section III-A in Gatelli 1994) and which is not caused*

*by volume scattering. In my opinion, another effect is volume decorrelation (Section III-B in Gatelli 1994) which occurs when different scatterers exist within the same resolution cell but at different viewing angles, hence they scatter with different inSAR phases which sum up coherently but random, therefore causing decorrelation.*

Response: We revised this statement and cite another reference (L43 to L45)

*line 75: "data from optical satellite sensors": which?*

Response: This is explained in detail in Section 2.2.

*line 96: "... bare ice appears on the surface (Fig. 1). The blue ice area (BIA)..." : Could you indicate consistently the location of the BIA on the map? The captions indicate the BIA with "B". Maybe, replace it with "BIA". or/and change "(Fig. 1)" do "(BIA in Fig. 1)". Could you add an arrow to the map indicating the wind direction and the location of its measurement? Possibly, also add the location of the stakes to the map as e.g., little black dots. As you are refering later to the ALE camp, could you also add it's location to the map?*

Response: We added the location of the camp and the met station. There are 50 stakes; showing these would spoil the figure. Locations of the stakes (in a figure and coordinates) and accumulation numbers are given by Rivera e al. (2014). We provide information on the wind direction in the text (L105). The BIA extent is variable.

*line 119-124: Could you add the transect, and possibly the thickness of the firn layers, to Fig. 1?*

Response: The location of the transect (L3) is shown in Fig. 8 of Uribe et al. (2014) and the radargram in Fig. 11. We explain the transect location in L122 to L125. Details on the profile location and refelcting layers can be looked up in the original paper.

*line 124 and 137: "raw SAR data". What do you mean with raw SAR data? Level 0 raw data or level 1b CoSSC data?*

Response: Raw data are Level 0 (notation added in L137). Co-registered Single-look Slant-range Complex (CoSSC) products are not raw data but are based on several processing steps.

*line 140: "the SAR amplitude, the backscattering coefficient": Are they not identical? Or do the authors mean different normalizations? Did the authors consider a backscatter dependence or normalization with respect to the local topography? If no radiometric terrain correction was applied, please justify and mention the expected error. See [D. Small, "Flattening Gamma: Radiometric Terrain Correction for SAR Imagery," in IEEE Transactions on Geoscience and Remote Sensing, vol. 49, no. 8, pp. 3081-3093, Aug. 2011, doi: https://doi.org/10.1109/TGRS.2011.2120616.]*

Response: SAR amplitude and backscatter coefficient are different raster output of the ITP. The SAR amplitude is represented in slant range and is uncalibrated radar brightness whereas the backscatter coefficient represents the power reflected from the ground, accounting for system noise and the local slope. All backscatter intensity data (backscatter images) generated for this study are based on absolute radiometric calibration and terrain-corrected geocoding using the high-resolution TanDEM-X global DEM, accounting for the local geocoded incidence angle, antenna beam pattern and correction for thermal noise. This delivers the actual sigma-0 value for each pixel. Terrain flattening performs backscatter normalization in order to reduce terrain-induced radiometric effects. This method is not applicable for studying the incidence angle dependence of backscatter.

*line 190: Even though stated in the introduction, I would repeat the information that snow pit measurements were done in Dec. 2016. This information is relevant for comparison of the snow pit data with the TanDEM-X data from different years.*

Response: "December 2016" added in L195.

*line 198 (also 371): "grain size": The next sentence indicates you measured "D_max"? Please specify. Maybe, add a reference, e.g. Mätzler et al (2002) "Relation between grain-size and correlation length of snow": https://doi.org/10.3189/172756502781831287 or the references to Colbeck 1990 or Armstrong 1993 therein.*

Response: Reference added in L205 and L664: Fierz et al., The International Classification for Seasonal Snow on the Ground, IACS 2009.

*line 209: "snow age following from different accumulation rates": Did you estimate accumulation rates or snow age from the snow pit measurements or from the accumulation stakes?*

Response: This refers mostly to the stakes and also to the ice core near Pit 3 and the snow pit site with a clear reference horizon (pit 5). Furthermore, the vicinity to the blue ice area (exposed to strong winds) implies lower accumulation rates than the area in the vicinity of the camp where the winds are less severe (this was a reason for selecting this site for the camp). This is clearly evident in the radargrams of Uribe et al. (2014). Relevant information is provided in L215 to L217.

*line 221: "accumulation rate near the ALE camp" Do you refer to snow pit P3 or to the accumulation stakes?*

Response: See comment above.

*line 388: To clarify that refraction has been considered, I suggest to write "backscatter simulations for $\theta_i = 40°$". Could you mention in line 338 how you obtained the refraction angle $\theta_r$ from snow density and $\theta_i$?*

Response: There is no need mentioning refraction explicitly. It is not possible to compute signal penetration and backscatter for a volume scattering medium without considering refraction. This is implicitly taken into account in the multi-layer models used for computing sigma-0.

*line 422: "The need for different parameter settings ... is an indication for structural anisotropy": Please rephrase. Neglecting a possibly(!) existing structural anisotropy (please define! see next comment) could be a possible reason, amongst others(!), why the backscatter model does not fit the observations.*

Response: The text on the related issue was revised and shifted to the Appendix.

*line 423-427: What do you mean with "structural anisotropy"? Do you mean the structural anisotropy of the microstructure (e.g. Leinss et al. "Modeling the evolution of the structural anisotropy of snow" The Cryosphere, 14, 51–75, 2020 https://doi.org/10.5194/tc-14-51-2020) or do you mean that horizontal layers with different density create a structural anisotropy of the snow pack (in the extreme case ice layers)? For such a layered snow pack I would expect a strongly angle-dependent backscatter dependency due to directional reflection at the layer-interfaces.*

Response: Radar signal propagation in polar firn is affected by structural anisotropy at different scales. The text on this issue was revised and shifted to the Appendix.

*Temperature gradient seems to be relatively low, but (Montagnat et a. (2020) "On the Birth of Structural and Crystallographic Fabric Signals in Polar Snow: A Case Study From the EastGRIP*

*Snowpack". Front. Earth Sci. 8:365. doi: https://doi.org/10.3389/feart.2020.00365) found for similar snow conditions a strong structural anisotropy of both, the c-axis and the microstructure. given the availability of VV and HH polarized acquisitions you could quickly check the copolar phase difference (Leinss, S., Löwe, et al.: Anisotropy of seasonal snow measured by polarimetric phase differences in radar time series, The Cryosphere, 10, 1771–1797, https://doi.org/10.5194/tc-10-1771-2016, 2016.) to estimate whether a strong structural anisotropy of the microstructure exists or whether you interpret the model-data discrepancy of the backscatter signal with incidence angle dependence through horizontal density variations as suggested by (Tan et al. 2017). In the latter case, I guess, without knowing the surface-roughness of each layer it seems impossible to model the precise incidence-angle dependent backscatter response of each layer. For discussing this effect, the work by Oh, Ulaby et al. could help: [Oh, Yisok, Kamal Sarabandi, and Fawwaz T. Ulaby. "An empirical model and an inversion technique for radar scattering from bare soil surfaces." IEEE transactions on Geoscience and Remote Sensing 30.2 (1992): 370-381.]*

Response: The reason for the increased sigma-0 towards near nadir incidence angles are very likely effects of (coherent and incoherent) scattering at interfaces and probably also interlayer interferences. We explain this in Sect. 6 (Discussion), L546 to L554 and also in the Appendix. We performed the proposed analysis of TanDEM-X VV-HH phase difference. However, such an analysis cannot provide definite information on the sources of the co-polar phase differences in a density stratified medium of semi-infinite depth. We report on this in Sect. 6 (Discussion), L555 to L581. The resulting CPD and HH-VV correlation coefficient show the same behaviour as observed on Alpine glaciers In Space Shuttle radar data, dominated by volume decorrelation in accumulation areas (Floricioiu and Rott, IEEE TGRS, Vol. 39(12), pp. 2634, 2001).

The Oh model is an empirical model developed for bare soil surfaces. It is not suitable for simulating the backscatter behaviour of interfaces in polar firn and explaining the scattering behaviour of a density-stratified medium.

*line 425: "angular gradients of the backscatter": Here and other places (especially also in the abstract) angular could refer to any direction or angle. Please be specific: I would rephrase that to "incidence angle dependence of the backscatter coefficient". Same for "angular difference".*

Response: This statement was removed. The backscatter topic is now treated in the Appendix.

*line 484: What is LGA? Please introduce abbreviation (level glacier area).*

Response: Now explained in L398to L400.

*line 484: Why did the authors exclude the BIA?*

Response: We exclude the BIA (a surface scattering target) in order to obtain statistics parameters for the volume scattering medium, including signal penetration, volumetric coherence, computed and observed elevation bias (Table 3).

*line 468: "coherence phase...is uniquely defined by the coherence magnitude" - That is only true for small penetration depth compared to the height of ambiguity which is given in your case, please clarify. See also Fischer et al. "Modeling Multifrequency Pol-InSAR Data From the Percolation Zone of the Greenland Ice Sheet" in IEEE Transactions on Geoscience and Remote Sensing, vol. 57, no. 4, pp. 1963-1976, April 2019, doi: https://doi.org/10.1109/TGRS.2018.2870301.*

Response: According to the model of Dall this statement is correct (see section 4 of Dall, 2007).

*Line 470-480: You describe two models (Dall 2007, Zebker 2000) to estimate the penetration depth from the coherence but both models provide different results. Could you provide reasons why you have chosen the model of Dall 2007 and why you think that this model describes better the relation between coherence and penetration depth?*

Response: We revised the related text put it into Sect. 3 providing basic information on coherence, signal penetration and elevation bias.

*Line 543-544: Please specify the wavenumber instead of using large / small.*

Response: We revised the related text (L579 to L585).

*Line 544: "The scene T2018 ... shows the smallest gradient" I see more a point cloud than a clear linear relation in Fig. 6b. Anyway, you write "as expected according to theory." Could you specify which theory you are referring to? Eq. 16?*

Response: We revised the related text (L582 to L589). The reference to theory is now treated in Sect. 6 (Discussion).

*Line 633-644: These lines reads like a general introduction rather than a discussion of your results. I would remove these lines.*

Response: Was removed.

*Line 656-663: Strongly shorten this section about the structural anisotropy to max. 1 sentence. You cannot start with "This can be explained." to finish with "However, such layers are absent... excluding anisotropy as a main explanation"*

Response: This text passage was removed. This topic is now treated in the Discussion section.

*Section 7: Please split this section into two sections, 7) Discussion and 8) conclusion. Line 689: I think here starts the conclusion.*

Response: Discussion and Conclusion are treated in separate sections.

*I think it is worth mentioning in the conclusion that despite its simplicity the approach from Dall 2007 provided reasonable results.*

Response: Thanks for the suggestion.

*Line 682-688: Please check references with cited content. They might have been flipped.*

Response: Thanks for noting this. Multilayer interactions are reported by Fischer at al. (2019a) and the tomographic analysis by Fischer at al. (2019b).

*Line 702-708: I don't see a point of citing the work of Parrella here because the authors have excluded the structural anisotropy as a reason for the observed incidence-angle dependent scattering.*

Response: We removed the reference to Parrella.

*technical corrections:*

*line 64: "DEM" is already defined in line 33.*

Response: Was skipped.

*line 77/78: "a well equipped field station" I guess, this is the "ALE camp" to which you are refering later. I think this is a better place to introduce the abbreviation "ALE" or "ALE camp". Currently,*

*mentioning Antarctic Logistics & Expeditions sounds a bit like company adverticement, but I guess the name is important to define the location of the ALE camp.*

Response: We skipped the first remark.

*Fig. 2: Please check whether the symbol size of grain shapes (and also fonts) is appropriate for the final layout.*

Response: We enlarged the symbol size.

*line 286: Here you introduce the abbreviation "IC2" which is sporadically used later. I think it's more consistent to define the appreviation at the very first location where IceSat-2 is introduced and use it then consistently. Or avoid the abbreviation if you prefer.*

Response: We deleted the abbreviation.

*Equation 4 (and other equations, variables and mathematical expressions): Only variables should be italic; functions "exp" and descriptive indices like "tot" should be upright. Latex: P_\text{tot}\text{exp}...*

Response: This equation is no more used.

*line 334: 37% reads a bit random. I guess you mean "attenuated to $e^{-1}$".*

Response: We replaced this by 1/e.

*line 485: level areas -> horizontal areas.*

Response: Horizontal is not the right word. The glacier surface is nowhere horizontal.

*line 494: "high values indicate large scattering elements": ... or steep slopes / layover / strong topographic variations.*

Response: This comment refers to the level glacier areas. Now explained in L407.

*Line 502 - referring to Fig. S2: Could the authors extend the color scale to cover the full range of backscatter differences? Possibly, clip the range at the 1 and 99% percentil to define the colorscale.*

Response: We selected this colour scale in order to optimize the visibility of the backscatter patterns over the LGA, the area of interest. For this reason we keep this scale. Sigma-0 values exceeding this range are locally apparent on steep slopes where the local incidence is at slanting angle or in the foreshortening.

*Fig. 4: Could the authors add the LGA mask to Fig. 4 for orientation?*

Response: We added the mask.

*line 507: "left of the camp" - do you mean ALE camp? As you are referring to the map, could you also indicate the location of the camp in Fig. 4 and 5?*

Response: This is near pit 4. We now refer to this location.

*line 509: "The angular difference... of ...-13.3.. is characteristic for" -> "The strong incidence angle dependence of the backscatter signal ... is characteristic for ... " (The current formulation implies that the specific values of the BIA are characteristic for ... )*

Response: Was reworded (L422).

*line 517: "has an also impact" -> "also has an impact"*

Response: Corrected.

---

## Author Response (AR2)

We would like to thank the referees for the careful review of the revised version of the manuscript and their positive comments. We incorporated the suggested changes, as explained below point-by-point. In addition, we implemented a few minor editorial changes (wording, clarifications).

*Referee comments are in italics,* our responses and explanations on implemented changes are in normal font. The line numbers ("L") in the response refer to the revised version of the manuscript with changes tracked.

**Authors Response to Referee 1**

*The authors should be commended for making this paper more readable for a broader audience. The structure of this revised manuscript is much improved, with clearer separation of the theory and assumptions that underpin the method and the application of it. The paper appears technically correct but would benefit from further clarifications to aid the reader and to ensure reproducibility. It would also be better to try to reduce the number of acronyms (perhaps in addition make a table in the appendix): it is jarring to try to recall what these are (particularly if they are used infrequently) and this interrupts the flow. Specific comments are:*

Response: We wish to thank the referee for his useful comments and the suggestions for changes addressing further clarifications and the improvement of readability. We incorporated the proposed changes as explained below. Besides, we checked the number of acronyms and their distribution in the text. We removed a few acronyms. Many acronyms refer to satellites, sensors and satellite products and are quite common. Some of the other acronyms require detailed explanations, better to be provided in the context rather than in a separate table. Therefore we refrain from lining up the acronyms in a table.

*Line 60: define SP-InSAR as single pass InSAR.*

Response: Defined (L 36).

*Line 127. It would really help to add a sentence here indicating why both the TDMgl and tiles are used in this study.*

Response: Explained, adding the following information (L132 to L135): "We use the TDMgl DEM for topographic corrections and geocoding because it provides full spatial coverage whereas the DEMs of individual tracks have gaps, depending on the observation geometry. The data from individual tracks are used for studying the impact of particular InSAR configurations on the coherence, backscatter signatures and the penetration bias."

*Line 140: 'geocoded rasters of the height error' – is this the Height Error Map from line 150 and comes directly from ITP?*

Response: This corresponds to the Height Error Map (reference added L147)

*Line 168: It would really help to add a sentence here to indicate IceSat / IceSat2 data are used to assess the temporal stability of the surface elevation and REMA is used to assess elevation bias because of its spatial coverage.*

Response: We added the following explanation (L179, 180): "We use the ICESat and ICESat-2 data primarily for assessing the temporal stability of the surface elevation." The information regarding the use of REMA is explained in L196.

*Line 214: 'At lower depth equi-temperature metamorphism takes over as dominant process for grain growth'. There are few measurements deeper than 2m and of these the grain type isn't showing strong*

*evidence for equitemperature metamorphism: these crystal types are also prolific higher in the pack which is dominated by temperature gradient metamorphism.*

Response: We skipped this sentence.

*Line 215: 'Estimates on accumulation rates are based on stake measurements (Rivera et al., 2014), the ice core between pit 2 and pit 3 (Hoffmann et al., 2020), and the summer melt crust in pit 5'. These accumulation rates and a brief description of how they were derived need to be presented, particularly as the discussion and even the abstract links the elevation bias to accumulation rates.*

Response: Sources and numbers for accumulation rates are addressed in a paragraph at the beginning of Section 2. For better visibility we added a separate sub-heading (2.1, L92) referring to surface mass balance (that corresponds to net accumulation in the areas of positive mass balance) and included further information on accumulation in this subsection.

*Figure 2: Colors in grain size legend still don't match the colors in the figure. E.g. on my computer smallest grain size in legend is #dde7f3, next smallest is #96b4f8. Lightest grain size in pit 5 is #b5bbde. I have not compared the hardness grayscale but would be hard-pushed to identify which category each would belong to. Would recommend switching hardness to different filling techniques (hatching). Please align the symbols used for crystal types with the International Classification for Seasonal Snow on the Ground – the symbol for 'faceted and rounded particles, closely packed' is not part of the classification and also is mismatched with the figure itself, which shows a closed form. You can download and use the fonts from https://cryosphericsciences.org/publications/snow-classification/.*

Response: Fig.2 was revised taking into account these comments.

*Line 247: 'The total snow mass down to 2.04 m amounts to 0.80 m w.e'. From Table 2, mean density for P4 is 403.75 kg / m3. For a depth of 2.04 m, this equates to a water equivalent of 2.04\*403.75/1000 m = 0.82 m. Does 0.80 m w.e. refer to just P4 or a mean of more than one pit?*

Response: This information refers to pit 4, as stated in the text. Number 4 of the hard layers (from the top) extends from 168 cm to 197 cm (with hardness R4) and 197 to 200 cm (R5), with rounding depth hoar below. We take the bottom of the hard layer (R5) as reference horizon, resulting in slightly revised numbers (L258): "Down to the depth of 2.0 m the P4 stratification shows four comparatively thick, hard layers with rounding depth hoar below. The total snow mass down to 2.0 m amounts to 0.81 m w.e."

*Line 311. Add in Dall 2007 reference again to show equation 11 is derived there not here.*

Response: For better explanation we added another equation (in response to a comment of referee 2) and another reference to Dall (2007) (L317 to L325).

*Line 338. 'For the elevation bias estimate derived from the volumetric coherence we use the notation hbInv'. Is this the same as hb in equation 11? If so, hbInv should be in equation 11. My understanding is that hbinv is expected to be the same as hb provided that hs is itself error-free. Unfortunately it is not because of potential temporal changes and the correction of REMA to CryoSat2 with its penetration correction and the following sections look at this. It would be helpful to provide some similar indication (or a correction of my understanding) to the reader to set the scene for sections 4.1.2 and 4.1.3.*

Response: Eqs. 10 to 12 (L320 to L328) are taken over from Dall (2007). These equations refer to the ideal case. In order to indicate the actual setting, we define in Section 4.1.1 notations for elevation differences derived from different topographic data and/or applying different registration procedures. We added a statement relating $H_{bInv}$ of Eq. 12 and $h_b$ of Eq. 13 (L354).

*Line 340 'On these surfaces the raw TDM DEMs show vertical offsets up to a few metres because for these data an absolute height calibration is not performed routinely'. Add reference into section 2.1 as this is presumably the 0.7 to 3m height error on line 158. If it is not, please add in an appropriate reference and explanation.*

Response: The values of 0.7 m to 3.0 m in the Raw DEMs are random errors at pixel scale due to phase noise which can be reduced by low-pass filtering (this is explained in L159 to L161). We added the following additional information in L161, L162: "The Height Error Map does neither account for the absolute height error (offset) in respect to a particular geodetic reference system nor for penetration related errors."

*Line 377: 'theTDMgl' -> 'the TDMgl'*

Response: Corrected, L394.

*Line 380: 'Subtracting the TDMgl offset of -6.76 m yields a dh value of -6.67 m due to signal penetration' would benefit from a rephrase: subtraction of a negative gives a positive.*

Response: We reformulated this statement (L397): "Subtracting the TDMgl offset of -6.76 m from the mean $\Delta h$ value (0.09 m) yields an elevation bias (dh) value of -6.67 m due to signal penetration.

*Line 407. Supplementary figures are out of order. S4 is referenced before S1.*

Response: We deleted here the reference to S4 (L425).

*Figure 3 and all relevant figures (especially Figure 5, difference a vs b and c vs d). Please put incidence angles in captions for ease of reading (rather than having to look them up in Table 1. The importance of incidence angle isn't evident before line 411, which is a long way past Table 1.*

Response: We added the incidence angle values in the captions of Figure 3, Figure 5, Figure 7; Figure 8; Figure S1:

*Line 417. This entire paragraph discusses supplementary figure S2. This is not supplementary and belongs in the same paper. Switching between files interrupts the flow of the paper.*

Response: We moved Fig. S2 to the main paper.

*Line 429. 'The coherence image of 6 May 2013 (Fig. 4) shows the lowest coherence (... < 0.7) on glacier sections with large signal penetration'. Where is the signal penetration shown? The following two sentences then contradict this. Coherence for BIA (surface reflection, no penetration) is 0.79 whereas the low accumulation areas (penetration) has coherence of 0.85 to 0.9. Please highlight the features of interest much more clearly.*

Response: Revised text (L454 to L458: "The coherence image of 6 May 2013 (Fig. 4) shows the lowest coherence on glacier sections with the largest elevation bias located on Driscoll Glacier and near the ALE camp ($\gamma_{tot}$ = 0.50 to 0.65). In the T2013 and T2014 (T2013/14) TDM images the coherence of the BIA is also comparatively low (mean $\gamma_{tot}$ = 0.79) because of thermal decorrelation due to the low SNR. In the low accumulation areas surrounding the BIA the $\sigma°$ values range from ..."

*Lines 433-437 discuss the role of incidence angle in interpreting the relationship between coherence and backscatter, supported by Figure S3. Figure S3 belongs in the main body of the paper.*

Response: We moved Fig. S3 to the main paper.

*Line 438. 'shows the expected variations in dependence of the height of ambiguity and incidence angle'. Please state here what the expected variations are as these are not obvious.*

Response: Information added (L465, 466): "..expected trend, i.e. i.e. decrease of $\gamma_{Vol}$ with increasing baseline (decreasing $H_a$) at a given incidence angle."

*Line 485. 'In order to check effects of the incidence angle the data derived from the T2013/14 and from the T2016/18 scenes are displayed separately' should come earlier, before Figure 5.*

Response: ...Moved to Sect. 4.3 (482, L483).

*Discussion. This isn't a criticism but merely noticeable that the first three paragraphs of the discussion cover material presented in the Appendix and supplementary material. It's a very interesting discussion but this gives the impression that the interesting material isn't in the main body of the paper. I'm not sure what to suggest here – perhaps no change needed.*

Response: Thanks for this positive feedback. In order to improve the visibility of relevant information we moved Figs. S2, S3 and S5 to the main paper.

*Line 555 – this paragraph appears without context. The structural anisotropy of Leinss et al is not the same as macroscale anisotropy (e.g. sastrugi) mentioned in the previous paragraph. This paragraph (if included) should be restructured to clarify the intended point.*

Response: We shifted this paragraph in order to follow the paragraph reporting on HH and VV properties for intensity, coherence and penetration. We start the paragraph on co-polarized phase difference and coherence with explaining the motivation. In the revision we provide also references on co-pol phase observations in dry polar firn, add some specific numbers for the LGAS and conclude a statement on the optional use of such data (L610 to L634).

*Line 574 – 'On the average' -> 'On average'*

Response: ..."on average " should be correct

*Line 580 – 'Both the dh and hbInv values indicate deeper penetration for T2018 compared to 2016, amounting on the LGA to 0.42 m, respectively 0.68 m' needs rephrasing. Does this mean e.g. 'Both the dh and hbInv values indicate deeper penetration for T2018 compared to 2016, amounting to 0.42 m for dh and 0.68 m for hbInv over the LGA'?*

Response: Reworded (L639, L640): 'Both dh and $h_{bInv}$ indicate a larger elevation bias (deeper penetration) for T2018 compared to T2016, amounting on the LGA to dh = -4.80 m for T2018 versus -4.38 m for T2016, respectively $h_{bInv}$ = -5.18 m for T2018 versus -4.40 m for T2016.

*Line 689. 'mean penetration depth which is deduced from the mean elevation difference' – how is this deduced? Does this mean that the mean penetration depth is assumed equal to the mean elevation difference?*

Response: ...Explained (L753): " ....which is deduced from the mean elevation difference (dh) between the T2013/14 scenes and the REMA assuming that the 2-way penetration depth is equal to the elevation bias."

*Line 697. 'smaller gain size' -> 'smaller grain size'.*

Response: ..Corrected

*Figure A1. Please explain how these simulations were carried out. It is not clear how the backscatter from below a layer is derived from total surface backscatter. How deep was the simulated profile? What was the lower boundary condition? Sufficient information must be given for these simulations to be recreatable.*

Response: This was explained in lines 675 – 676. We added further explanation added (L735 to L738): "We use $\tau$ as tuning parameter in order to match the average observed and computed total backscatter intensity at the individual sites. .....   The computations were performed down to 20m

depth. Contributions to total backscatter from the layers below are negligible because of the dense medium effect and the attenuation in the layers above."

*Figure A1. Not for this paper, but it would be really interesting to try to understand how the relative differences between the uniform volume approach and the layered scattering model approach translates to penetration bias. 3dB (line 704) is a large error. The error for pits 2-5 should be reported here rather than simply stating 'The RT simulations are able to reproduce the observed total backscatter intensity at snow pit sites 2 to 5'.*

Response: Further explanation added (L766 to L769): "The RT simulations for $\theta_i = 40°$, using $\tau$ as tuning parameter, reproduce exactly the observed mean total backscatter intensity of the corresponding T2013/14 data at snow pit sites 2 to 5. For pit 1 the simulations for $\theta_i = 40°$ yield an underestimation of 3 dB, even when assuming consistently maximum stickiness ($\tau = 0.1$), ......."

(For snow pit sites 2 to 5 (40°) the simulations are matching the observations with 0.1 dB)

*Line 707. 'Such an angular difference is typical for density-layered firn' – does this refer to the 1.3 or 5.9 dB?*

Response: Explained (L772): "Large incidence angle dependence of backscatter is typical for density-layered firn."

**Authors Response to Referee 2**

*General Comment*

*The authors thoroughly revised the manuscript. The revision is better balanced and provides a more objective interpretation of the numerous observations and results. The results are valuable for interpretation of radar penetration, coherence and backscatter response over ice sheets and glaciers. While the authors present promising results using the relatively simple Dall model to correct for radar penetration, they also address problems in simulating the backscatter response from stratified snow. Overall, I can recommend publication of the manuscript in TC. Below are some technical corrections which I encourage the authors to consider before submitting the final manuscript*

Response: We wish to thank the referee for his useful comments and the suggestions regarding technical corrections or clarifications. We have incorporated implemented the proposed changes as explained below.

*Technical corrections:*

*line 146: What do you mean by "absolute radiometric calibration and terrain-corrected geocoding" ? Do you mean, the backscatter signal was radiometrically terrain corrected? gamma_0 or sigma_0? Or was it only geometrically terrain corrected (orthorectified) as written above ("geocoded rasters of ... SAR amplitude")*

Response: For clarification we replaced this sentence by (L153 to L155): "The backscatter intensity images show maps of the normalized radar cross section $\sigma°$. For the computation of $\sigma°$ effects of topography were taken into account for antenna pattern removal and for defining the actual size of the local scattering area." The notation "normalized radar cross section" is a standard term for absolutely calibrated backscatter values referring to the unit surface area.

*Figure 2: Comparing the colorbar for grain size with the color-coded column "GS" appears as different color scales were used. Please correct.*

Response: The colour scale was revised for better discrimination of the grain size and the hardness classes.

*line 306: "Normalizing the coherence by the interferometric phase of the volume surface": Could you provide an equation make this sentence more comprehensible? Do you divide two complex numbers here.*

Response: For better explanation we revised the text and added another equation (L317 to L325).

*line 310: "As according to this relation [Eq. 10] the coherence phase, <gamma, is uniquely defined by the coherence magnitude": Eq. 10 does not illustrate that the coherence phase is uniquely defined by the coherence magnitude.*

Response: See response to comment above. Eq. 11 (added) shows this relation.

*line 397: "total normalized coherence": could you provide a reference to an equation which defines the "total normalized coherence"?*

Response: "Normalized coherence" is a common term for the magnitude of the complex interferometric correlation coefficient (often simply called "coherence"). The total complex correlation coefficient is specified in equation 1. Revised text (L415, L416):" Fig. 3 shows an image of the magnitude of the complex interferometric correlation coefficient (the total normalized coherence) ..."

*line 399: "excluding (...) and slopes smaller than 5° inclination": I guess you mean larger than 5°?*

Response: Thanks for noting this. Corrected (L418).

*line 422: "Large angular gradients" - See original comment by Referee #2: "angular gradients of the backscatter" Here and other places (...) angular could refer to any direction or angle. Please be specific: I would rephrase that to "incidence angle dependence". Same for "Angular difference" in Fig. S2. I suggest to delete "angular" in the caption as the different incidence angles are already given in parenthesis*

Response: We replaced the word "angular gradients" in L442, L572, L772 and deleted the word "angular" in the caption of Fig. S2 (now Fig. 5).

---

## Author Response (AR3)

Manuscript tc-2020-380,  Authors response to Editor:

We would like to thank the editor efforts. We implemented the minor changes as proposed. In particular, we replaced the term LGA (level glacier area) by AoI (area of interest) throughout the manuscript and in the Supplement. In line 409 to line 412 we included related information.